# Regularized Frank-Wolfe for Dense CRFs: Generalizing Mean Field and Beyond

**Đ.Khuê Lê-Huu**          **Karteek Alahari**

Univ. Grenoble Alpes, Inria, CNRS, Grenoble INP, LJK

38000 Grenoble, France

{khue.le,karteek.alahari}@inria.fr

## Abstract

We introduce *regularized Frank-Wolfe*, a general and effective algorithm for inference and learning of dense conditional random fields (CRFs). The algorithm optimizes a nonconvex continuous relaxation of the CRF inference problem using vanilla Frank-Wolfe with approximate updates, which are equivalent to minimizing a regularized energy function. Our proposed method is a generalization of existing algorithms such as mean field or concave-convex procedure. This perspective not only offers a unified analysis of these algorithms, but also allows an easy way of exploring different variants that potentially yield better performance. We illustrate this in our empirical results on standard semantic segmentation datasets, where several instantiations of our regularized Frank-Wolfe outperform mean field inference, both as a standalone component and as an end-to-end trainable layer in a neural network. We also show that dense CRFs, coupled with our new algorithms, produce significant improvements over strong CNN baselines.

## 1 Introduction

Fully-connected or *dense* conditional random fields (CRFs) [34]—combined with strong pixel-level classifiers such as a convolutional neural network (CNN) [25, 42]—have been a highly-successful paradigm for semantic segmentation. Top-performing systems on the PASCAL VOC benchmark [22] used to include a CRF as either a post-processing step [13, 14, 15, 20, 43, 44, 45] or a trainable component [4, 49, 64, 68, 73, 76]. However, as CNNs got stronger, the improvements that CRFs brought decreased over time, and as a result they fell out of favor since 2017 [45].

In this paper, we revisit dense CRFs with two contributions. First, on the theoretical side, we propose *regularized Frank-Wolfe*, a new class of algorithms for inference and learning of CRFs that perform better than the popular *mean field* [34, 35, 58]—the method of choice in the aforementioned works. Regularized Frank-Wolfe optimizes a nonconvex continuous relaxation of the CRF inference problem (§2) by performing approximate conditional-gradient updates (§3.1), which is equivalent to minimizing a regularized energy using the generalized Frank-Wolfe method [53] (§3.2). Several of its instantiations lead to new algorithms that have not been studied before in the MAP inference literature (§3.3). Moreover, we show that it also includes several existing methods, including mean field and the concave-convex procedure [75], as special cases (§3.4). This generalized perspective allows a unified analysis of all these old and new algorithms in a single framework (§4). In particular, we show that they achieve a sublinear rate of convergence $\mathcal{O}(1/\sqrt{k})$ for suitable stepsize schemes, and in certain cases (such as strongly-convex regularizer or concave energy) this can be improved to $\mathcal{O}(1/k)$ (§4.1). Furthermore, we provide a tightness analysis for the resulting nonconvex relaxation of the regularized energy, recovering some existing tightness results [10, 41, 61] as special cases (§4.2). The proposed algorithms are easy to implement, converge quickly in practice, and have (sub)differentiable iterates. Such properties are important for successful *gradient-based learning* via backpropagation [47, 62].

35th Conference on Neural Information Processing Systems (NeurIPS 2021).

Our second contribution lies on the practical side. In addition to mean field and regularized Frank-Wolfe variants, we re-implement several existing first-order inference methods [39, 41, 48]—those that are amenable to gradient-based learning—for comparison. Remarkably, we find that dense CRFs can still achieve important improvements over the strong DeepLabv3+ [17] CNN model for all these solvers (§5). In particular, our best variant of regularized Frank-Wolfe achieves a mean intersection-over-union (mIoU) score of 88.0 on the PASCAL VOC test set (§5.3), improving over DeepLabv3+. We hope that these encouraging results could attract interest from the community in considering dense CRFs (again) for tasks such as semantic segmentation. Our source code is made publicly available under the GNU general public license for this purpose.[1]

## 2   Background

### 2.1   Inference in CRFs

Let $\mathbf{s} \in \mathcal{S}_1 \times \cdots \times \mathcal{S}_n$ denote an assignment to $n$ discrete random variables $S_1, \ldots, S_n$, where each variable $S_i$ takes values in a finite set of states (or *labels*) $\mathcal{S}_i$. Let $\mathcal{G} = (\mathcal{V}, \mathcal{E})$ be a graph of $n$ nodes ($\mathcal{V} = \{1, 2, \ldots, n\}$). A Markov random field (MRF) defined by $\mathcal{G}$ encodes a family of joint distributions that can be factorized as follows, where $\phi_i : \mathcal{S}_i \to \mathbb{R}_+$ and $\phi_{ij} : \mathcal{S}_i \times \mathcal{S}_j \to \mathbb{R}_+$ are the so-called *unary* and *pairwise* (respectively) potential functions, and $Z$ is a normalization factor:

$$p(\mathbf{s}) = \frac{1}{Z} \prod_{i \in \mathcal{V}} \phi_i(s_i) \prod_{ij \in \mathcal{E}} \phi_{ij}(s_i, s_j). \tag{1}$$

Note that (1) can also include conditional distributions, i.e., $p(\mathbf{s} \mid \mathbf{o})$ with observed variables $\mathbf{o}$. In this case the potentials may also depend on $\mathbf{o}$, e.g., $\phi_i(s_i; \mathbf{o})$, and this model is referred to as conditional random field (CRF) [37]. We will present later (§5.1) such a model for image segmentation. In the following, we use MRF and CRF interchangeably. We assume further that all nodes have the same set of labels: $\mathcal{S}_i = \mathcal{S} \ \forall i$, with cardinality $d = |\mathcal{S}|$. It is convenient to express $p(\mathbf{s})$ as $\frac{1}{Z} \exp(-e(\mathbf{s}))$, where the so-called *energy* $e(\mathbf{s})$ is defined as

$$e(\mathbf{s}) = \sum_{i \in \mathcal{V}} \theta_i(s_i) + \sum_{ij \in \mathcal{E}} \theta_{ij}(s_i, s_j), \quad \text{with } \theta_i(s_i) = -\log \phi_i(s_i) \text{ (idem for } \theta_{ij}). \tag{2}$$

The task of maximum a posteriori (MAP) inference consists in finding the most probable joint assignment, also known as *energy minimization* (which is NP-Hard in general [65]):

$$\mathbf{s}^* = \underset{\mathbf{s} \in \mathcal{S}^n}{\operatorname{argmax}} \, p(\mathbf{s}) = \underset{\mathbf{s} \in \mathcal{S}^n}{\operatorname{argmin}} \, e(\mathbf{s}). \tag{3}$$

### 2.2   Continuous relaxation of MAP inference

Let $x_{is}$ be a binary variable such that $x_{is} = 1$ iff label $s$ is assigned to node $i$. Then, $\mathbf{x}_i = (x_{is})_{s \in \mathcal{S}} \in \{0, 1\}^d$ denotes the one-hot vector for node $i$. Let $\mathbf{x} \in \{0, 1\}^{nd}$ be the concatenation of all $\mathbf{x}_i$, and let $\boldsymbol{\theta}_i = (\theta_i(s))_{s \in \mathcal{S}} \in \mathbb{R}^d, \boldsymbol{\Theta}_{ij} = (\theta_{ij}(s, t))_{t \in \mathcal{S}}^{s \in \mathcal{S}} \in \mathbb{R}^{d \times d}$. The energy (2) is then

$$E(\mathbf{x}; \boldsymbol{\theta}) = \sum_{i \in \mathcal{V}} \boldsymbol{\theta}_i^\top \mathbf{x}_i + \sum_{ij \in \mathcal{E}} \mathbf{x}_i^\top \boldsymbol{\Theta}_{ij} \mathbf{x}_j, \tag{4}$$

where $\boldsymbol{\theta}$ is a parameter vector composed of all $\boldsymbol{\theta}_i$ and $\boldsymbol{\Theta}_{ij}$. The MAP inference problem (3) transforms to minimizing $E(\mathbf{x}; \boldsymbol{\theta})$ over the new variables $\mathbf{x}$. A natural approach is to relax the binary constraint and solve the continuous relaxation $\min_{\mathbf{x} \in \mathcal{X}} E(\mathbf{x}; \boldsymbol{\theta})$ with

$$\mathcal{X} = \left\{ \mathbf{x} \in \mathbb{R}^{nd} : \mathbf{x} \geq \mathbf{0}, \mathbf{1}^\top \mathbf{x}_i = 1 \ \forall i \in \mathcal{V} \right\}. \tag{5}$$

This continuous relaxation is known to be *tight* [10, 41, 61]. For convenience, we represent the unary potentials as a vector $\mathbf{u}(\boldsymbol{\theta}) = (\boldsymbol{\theta}_i)_{i \in \mathcal{V}} \in \mathbb{R}^{nd}$, and the pairwise potentials as a symmetric $n \times n$ block matrix $\mathbf{P}(\boldsymbol{\theta}) \in \mathbb{R}^{nd \times nd}$, where the $(i, j)$ block is $\boldsymbol{\Theta}_{ij}$. The problem is reduced to

$$\min_{\mathbf{x} \in \mathcal{X}} E(\mathbf{x}; \boldsymbol{\theta}) \triangleq \frac{1}{2} \mathbf{x}^\top \mathbf{P}(\boldsymbol{\theta}) \mathbf{x} + \mathbf{u}(\boldsymbol{\theta})^\top \mathbf{x}. \tag{6}$$

---

[1] https://github.com/netw0rkf10w/CRF

The reason we have made $\boldsymbol{\theta}$ explicit in (4) and (6) is to provide more clarity when discussing the differentiability of their solutions for the *learning* task. Note that an optimal solution $\mathbf{x}^*$ to (4) and (6) is a function of $\boldsymbol{\theta}$, and thus should be written as $\mathbf{x}^*(\boldsymbol{\theta})$. Therefore, when we say a solution is differentiable, it is understood that it is so with respect to $\boldsymbol{\theta}$. When there is no ambiguity, we omit $\boldsymbol{\theta}$ and write simply $E(\mathbf{x})$, $\mathbf{P}$, and $\mathbf{u}$. We will be interested in problem (6) in the rest of the paper, though it should be noted that our method also applies to the so-called linear programming (LP) relaxation, which takes the form $\min_{\mathbf{x} \in \mathcal{X}_{\mathrm{LP}}} E_{\mathrm{LP}}(\mathbf{x}; \boldsymbol{\theta})$, where $E_{\mathrm{LP}}(\mathbf{x}; \boldsymbol{\theta}) = \boldsymbol{\theta}^\top \mathbf{x}$ and $\mathcal{X}_{\mathrm{LP}}$ is the so-called *local polytope* [72]. We refer to Appendix A for the details.

## 2.3 Vanilla Frank-Wolfe algorithm for MAP inference

Since the continuous energy is differentiable, it is natural to apply first-order methods such as Frank-Wolfe [23] to solving (6) [41]. Starting from a feasible $\mathbf{x}^0 \in \mathcal{X}$, Frank-Wolfe approximately solves (6) by iterating the following steps, where $\alpha_k \in [0, 1]$ follows some stepsize scheme:

$$\mathbf{p}^k \in \underset{\mathbf{p} \in \mathcal{X}}{\operatorname{argmin}} \left\langle \nabla E(\mathbf{x}^k), \mathbf{p} \right\rangle, \qquad \mathbf{x}^{k+1} = \mathbf{x}^k + \alpha_k(\mathbf{p}^k - \mathbf{x}^k). \tag{7}$$

The same idea has also been successfully applied to other types of continuous relaxations of (6), such as linear programming (LP) [52] or convex quadratic programming (QP) [21, 61].

# 3 Regularized Frank-Wolfe for inference

In this section, we introduce the proposed regularized Frank-Wolfe as a general class of algorithms for MAP inference. The motivation and general framework are presented in §3.1 and §3.2. We show that several new inference algorithms can be obtained using different regularizers (§3.3). Moreover, regularized Frank-Wolfe also includes a number of existing algorithms as special cases (§3.4).

## 3.1 A smoothing perspective

We have seen in §2.3 the (vanilla) Frank-Wolfe method for solving MAP inference. Unfortunately, from a *learning* perspective, this algorithm is problematic. Indeed, CRF learning with stochastic gradient descent (SGD) is typically done by backpropagating through the optimization steps [35, 63, 76], but the iterate $\mathbf{p}^k$ in (7) is piecewise constant and thus its gradient (w.r.t. $\boldsymbol{\theta}$) is zero almost everywhere (§C.1), which makes learning not possible.[2] This issue will be illustrated later in the experiments. A potential solution is to add a (typically strongly-)convex regularization term:

$$\mathbf{p}^k \in \underset{\mathbf{p} \in \mathcal{X}}{\operatorname{argmin}} \left\{ \left\langle \nabla E(\mathbf{x}^k), \mathbf{p} \right\rangle + r(\mathbf{p}) \right\}, \qquad \mathbf{x}^{k+1} = \mathbf{x}^k + \alpha_k(\mathbf{p}^k - \mathbf{x}^k). \tag{8}$$

With appropriately chosen regularizers, (8) becomes suitable for gradient-based learning.

The technique of approximating an update step by a regularized one is quite standard in the optimization literature. For example, the classical proximal gradient method [48] can be interpreted the same way. Our update (8) is inspired by Nesterov's smoothing [56], and thus is similar in spirit to its many applications [31, 57, 66]. In the next section, we present a different, more general perspective to view (8), which offers more flexibility in designing new algorithms, in making connections to existing ones, and in analyzing their theoretical properties in a unified manner.

## 3.2 A regularized energy perspective

Instead of approximating the local updates of a *given algorithm* (in this case: vanilla Frank-Wolfe) to minimize the *same* objective function, one may choose to keep the algorithm, and approximate instead the objective. At first glance, however, this idea does not seem to be a good one. Indeed, if we replace $E$ by some function $E'$ and apply vanilla Frank-Wolfe, the updates (7) remains piecewise constant, thus we have the same zero-gradient issue. It turns out that, if we choose a more appropriate "given algorithm", this idea can work. Such a choice is the *generalized Frank-Wolfe* algorithm [5, 12, 53].

---

[2]Note that backpropagation typically requires the operations to be differentiable (at least) almost everywhere, which $\mathbf{p}^k$ satisfies. Therefore, the issue here does not lie in *differentiability*, but in the resulting *zero* gradients. Blackbox differentiation [60] can deal with this scenario, but it is limited to LP relaxations.

Consider the following problem, where $f : \mathbb{R}^m \to \mathbb{R} \cup \{+\infty\}$ is differentiable but possibly nonconvex, and $g : \mathbb{R}^m \to \mathbb{R} \cup \{+\infty\}$ is proper, closed, and convex but possibly non-differentiable:

$$\min_{\mathbf{x}} F(\mathbf{x}) \triangleq f(\mathbf{x}) + g(\mathbf{x}). \tag{9}$$

Generalized Frank-Wolfe solves (9) by iterating

$$\mathbf{p}^k \in \operatorname*{argmin}_{\mathbf{p}} \left\{ \langle \nabla f(\mathbf{x}^k), \mathbf{p} \rangle + g(\mathbf{p}) \right\}, \qquad \mathbf{x}^{k+1} = \mathbf{x}^k + \alpha_k (\mathbf{p}^k - \mathbf{x}^k). \tag{10}$$

If $g$ is the indicator function $\delta_{\mathcal{X}}$ of $\mathcal{X}$ (i.e., $\delta_{\mathcal{X}}(\mathbf{x}) = 0$ if $\mathbf{x} \in \mathcal{X}$ and $\delta_{\mathcal{X}}(\mathbf{x}) = +\infty$ otherwise), then the algorithm clearly reduces to vanilla Frank-Wolfe (7) for $\min_{\mathbf{x} \in \mathcal{X}} f(\mathbf{x})$. Therefore, generalized Frank-Wolfe applied to (6) with $f = E$ and $g = \delta_{\mathcal{X}}$ will yield exactly the same updates (7). Now let us apply this algorithm to an approximate objective $E_r(\mathbf{x}) = E(\mathbf{x}) + r(\mathbf{x})$ for some function $r$. Choosing $f = E$ and $g = r + \delta_{\mathcal{X}}$, it is straightforward that (10) reduces to (8). Therefore, we have recovered the same algorithm as in §3.1, but this time through different machinery.

This framework offers a great flexibility as one can choose $f$ and $g$ in many different ways to obtain new algorithms. The only conditions are $f$ being differentiable and $g$ being convex, so that the subproblem in (10) is well-defined and globally solvable.[3] For example, instead of choosing $f = E$ and $g = r + \delta_{\mathcal{X}}$ as above, one can choose $f(\mathbf{x}) = \frac{1}{2} \mathbf{x}^\top \mathbf{P} \mathbf{x}$ and $g(\mathbf{x}) = \mathbf{u}^\top \mathbf{x} + r(\mathbf{x}) + \delta_{\mathcal{X}}(\mathbf{x})$. We will recover later in §3.4 some existing algorithms (as special cases) through this kind of decomposition. Finally, we present Algorithm 1 for (approximately) solving MAP inference (6).

---

**Algorithm 1** Generic regularized Frank-Wolfe for (approximately) solving MAP inference (6).

---

1: Choose a regularizer $r$ such that there exist $f$ (differentiable) and $g$ (convex) satisfying $f + g = E + r + \delta_{\mathcal{X}}$. Typically (but not necessarily) $r$ is convex on $\mathcal{X}$ and is constant on $\mathcal{X} \cap \{0, 1\}^{nd}$.
2: Initialization: $k \leftarrow 0$, $\mathbf{x}^0 \in \mathcal{X}$, number of iterations $N$.
3: Compute $\mathbf{p}^k \in \operatorname{argmin}_{\mathbf{p}} \left\{ \langle \nabla f(\mathbf{x}^k), \mathbf{p} \rangle + g(\mathbf{p}) \right\}$ and compute the stepsize $\alpha_k$.
4: Update $\mathbf{x}^{k+1} = \mathbf{x}^k + \alpha_k (\mathbf{p}^k - \mathbf{x}^k)$. Let $k \leftarrow k + 1$ and go to Step 3 until $k = N$.
5: Rounding: convert $\mathbf{x}$ to a discrete solution and return.

---

While the choice of $(r, f, g)$ can be highly flexible, it would make little sense to optimize a function that has nothing to do with the original objective (i.e., the discrete energy). Let $\overline{\mathcal{X}} = \mathcal{X} \cap \{0, 1\}^{nd}$ denote the discrete domain of our problem. If we choose $r$ such that it is constant on $\overline{\mathcal{X}}$ (as suggested in Step 1 above), then minimizing $E$ on $\overline{\mathcal{X}}$ is equivalent to minimizing $E + r$ on $\overline{\mathcal{X}}$, and thus Algorithm 1 actually solves the continuous relaxation of a (different) discrete problem that is equivalent to MAP inference. Further discussion on this matter, as well as on the rounding Step 5, are deferred until §4.2.

Finally, we should note that adding a strongly-convex regularizer is not new in the MAP inference literature [29, 31, 52, 67, 69]. In particular, some previous work even applied (vanilla) Frank-Wolfe to optimizing such regularized energy [52, 67, 69]. All these algorithms, however, suffer from the aforementioned zero-gradient issue, as already explained in the beginning of this section.

## 3.3 Particular instantiations

The previous section presents regularized Frank-Wolfe as a general algorithm for inference. We now discuss concrete examples of its instantiations. To the best of our knowledge, *all the algorithms presented in this section are new and have not been studied previously in the MAP inference literature*. In particular, despite some similarities with proximal gradient [48] and mirror descent [8, 55], our following euclidean and entropic variants are actually different from these methods.[4]

**Euclidean Frank-Wolfe** Perhaps the most natural choice is $\ell_2$ regularization. In Algorithm 1, let us choose $f(\mathbf{x}) = E(\mathbf{x})$ and $r(\mathbf{x}) = \frac{\lambda}{2} \|\mathbf{x}\|_2^2$, where $\lambda > 0$ is a regularization weight. Let $\Pi_{\mathcal{X}}(\mathbf{v})$ be the projection of a vector $\mathbf{v}$ onto $\mathcal{X}$. It can be shown (§D.1) that Step 3 in Algorithm 1 becomes

$$\mathbf{p}^k = \operatorname*{argmin}_{\mathbf{p} \in \mathcal{X}} \left\{ \langle \mathbf{P}\mathbf{x}^k + \mathbf{u}, \mathbf{p} \rangle + \frac{\lambda}{2} \|\mathbf{p}\|_2^2 \right\} = \Pi_{\mathcal{X}} \left( -\frac{1}{\lambda} (\mathbf{P}\mathbf{x}^k + \mathbf{u}) \right) \quad \forall k \geq 0. \tag{11}$$

---

[3]Further mild conditions are required for convergence (§4.1).

[4]In proximal gradient and mirror descent, the current iterate is constrained to stay close to the previous one, while this is not the case in our method. See Appendix D for the details.

**Entropic Frank-Wolfe**  In Algorithm 1, let us choose $f(\mathbf{x}) = E(\mathbf{x})$ and $r(\mathbf{x}) = -\lambda H(\mathbf{x})$, where $\lambda > 0$ is a regularization weight and $H(\mathbf{x}) = -\sum_{i \in \mathcal{V}} \sum_{s \in \mathcal{S}} x_{is} \log x_{is}$ is the entropy of $\mathbf{x}$ over $\mathcal{X}$. It can be shown (§D.2) that Step 3 in Algorithm 1 becomes

$$\mathbf{p}^k = \underset{\mathbf{p} \in \mathcal{X}}{\operatorname{argmin}} \left\{ \langle \mathbf{P}\mathbf{x}^k + \mathbf{u}, \mathbf{p} \rangle - \lambda H(\mathbf{p}) \right\} = \operatorname{softmax}\left( -\frac{1}{\lambda}(\mathbf{P}\mathbf{x}^k + \mathbf{u}) \right) \quad \forall k \geq 0, \qquad (12)$$

where $\mathbf{v} = \operatorname{softmax}(\mathbf{x})$ with $\mathbf{x} \in \mathbb{R}^{nd}$ means $\mathbf{v} \in \mathbb{R}^{nd}$ and $v_{is} = \frac{\exp(x_{is})}{\sum_{t \in \mathcal{S}} \exp(x_{it})}$ $\forall i \in \mathcal{V}, \forall s \in \mathcal{S}$. The resulting algorithm has a tight connection with (parallel) mean field [34, 35] (discussed in §3.4).

**Other variants**  One can consider more sophisticated regularizers, e.g., a weighted combination of $\ell_2$ norm and entropy. Other options include the many different regularizers that have been used in diverse machine learning applications, such as $\ell_p$ norm [57], lasso variants [57], or binary entropy [3]. Although these variants also lead to new MAP inference algorithms, their implementations are more sophisticated since their subproblems (10) require numerical solutions as no closed form ones exist.

### 3.4  Recovering existing algorithms as special cases

In addition to the above new algorithms, regularized Frank-Wolfe also includes several existing ones as special cases. We present some of them below and refer to Appendix A for further details.

**Mean field**  This is a special case of the above Entropic Frank-Wolfe. Indeed, if we choose $\lambda = 1$ in (12) and a constant stepsize $\alpha_k = 1$ $\forall k \geq 0$ in Algorithm 1, then it is straightforward that this algorithm is reduced to the following update step, where $\mathcal{N}_i$ is the set of neighbors of node $i$:

$$\mathbf{x}^{k+1} = \operatorname{softmax}(-\mathbf{P}\mathbf{x}^k - \mathbf{u}) \iff x_{is}^{k+1} = \tfrac{1}{Z_i} \exp\left( -\theta_i(s) - \sum_{j \in \mathcal{N}_i} \sum_{t \in \mathcal{S}} \theta_{ij}(s,t) x_{jt}^k \right) \forall i \in \mathcal{V}, s \in \mathcal{S}.$$

This is precisely a (parallel) mean field update [34, 35]. To conclude, parallel mean field is an instance of Entropic Frank-Wolfe with unit regularization weight and unit stepsize. Interestingly, the update (12) is the well-known softmax function with *temperature* in the deep learning literature [28]. One could have easily come up with such a simple extension of mean field by adding a temperature to softmax (yet surprisingly this has not been tried before), but here we have provided a principled way to achieve that. As shown later in the experiments, with suitable $\lambda$, this extension yields much better results than vanilla mean field. Finally, we should note that the tight connection between mean field and first-order methods has been noticed before. Krähenbühl and Koltun [35] proposed several mean-field-type variants based on the concave-convex procedure [75], while closely similar variants can also be obtained through proximal gradient [2, 6], but unlike our generalized algorithm, these algorithms cannot recover *exactly* the original mean field of Krähenbühl and Koltun [34].

**Concave-convex procedure**  CCCP [75] solves (9), assuming $f$ is concave and $g$ is convex, by updating $\mathbf{x}^{k+1}$ as a solution to $-\nabla f(\mathbf{x}^k) \in \partial g(\mathbf{x}^{k+1})$,[5] which is precisely (10) with stepsize $\alpha_k = 1$. We conclude that CCCP is a special case of generalized Frank-Wolfe with $f$ concave and unit stepsize. As a result, many existing CCCP-based inference algorithms [21, 35] can be seen as special cases of regularized Frank-Wolfe. For example, the ones presented by Desmaison et al. [21] are instantiations of the proposed algorithm with either $f(\mathbf{x}) = -\mathbf{x}^\top \operatorname{diag}(\mathbf{c})\mathbf{x}$ and $r(\mathbf{x}) = E(\mathbf{x}) + \mathbf{x}^\top \operatorname{diag}(\mathbf{c})\mathbf{x}$ (where $\mathbf{c} \in \mathbb{R}^{nd}$ is large enough so that $r(\mathbf{x})$ is convex), or $f(\mathbf{x}) = \mathbf{x}^\top(\mathbf{P} - \mathbf{C})\mathbf{x}$ and $r(\mathbf{x}) = \mathbf{u}^\top\mathbf{x} + \mathbf{x}^\top\mathbf{C}\mathbf{x}$ (where $\mathbf{C}$ is some matrix such that $f$ is concave and $r$ is convex). Note that in these instantiations, Step 3 in Algorithm 1 requires an iterative (numerical) solution. Finally, all the algorithms presented by Krähenbühl and Koltun [35] are also instantiations of the proposed method because they are based on CCCP. We refer to Appendix A for further details.

**Vanilla Frank-Wolfe**  This is trivially a special case of regularized Frank-Wolfe and we briefly discuss it for completeness. Choosing $f(\mathbf{x}) = E(\mathbf{x})$ and $r(\mathbf{x}) = 0$ we obtain the algorithm by Lê-Huu and Paragios [41]. Likewise, the one by Desmaison et al. [21] corresponds to $f(\mathbf{x}) = E(\mathbf{x}) - \mathbf{c}^\top\mathbf{x} + \mathbf{x}^\top \operatorname{diag}(\mathbf{c})\mathbf{x}$ and $r(\mathbf{x}) = 0$, where $\mathbf{c} \in \mathbb{R}^{nd}$ is large enough for $f$ to be convex. In addition, we can also recover existing LP-based algorithms by choosing $\mathcal{X} = \mathcal{X}_{\mathrm{LP}}, r(\mathbf{x}) = 0$, and $f(\mathbf{x}) = E_{\mathrm{LP}}(\mathbf{x}) + R(\mathbf{x})$ with suitable $R(\mathbf{x})$. Indeed, the one by Meshi et al. [52] takes $R(\mathbf{x})$ as the squared $\ell_2$-norm of linear constraints, while the ones by Sontag and Jaakkola [67] and Tang et al. [69] correspond to $R(\mathbf{x})$ being an entropy approximation and its generalization, respectively (see §A).

---

[5]In the original CCCP [75], $g$ is differentiable, thus the update becomes $-\nabla f(\mathbf{x}^k) = \nabla g(\mathbf{x}^{k+1})$.

## 4 Theoretical analysis

### 4.1 Convergence

We provide a convergence analysis for the generalized Frank-Wolfe algorithm, and the results for CRF inference special cases will then follow as a consequence. Convergence of vanilla Frank-Wolfe has been well studied in the literature [24, 30, 36, 38]. For generalized Frank-Wolfe, different analyses exist for the case where both $f$ and $g$ in (9) are convex [5, 26, 51, 74]. We are particularly interested in the general case where $f$ is **nonconvex**,[6] as the CRF energy is often highly so in practice. Mine and Fukushima [53] (and subsequently Bredies et al. [12]) proved the global convergence of the algorithm under mild conditions, though no rate of convergence was given. Recently, Beck [7] obtained an $\mathcal{O}(1/\sqrt{k})$ rate of convergence for convex $g$ under adaptive or line-search stepsizes. We extend their analysis with several contributions. First, we include the case where $g$ is strongly convex, which is important as our main variants for inference (e.g., mean field or $\ell_2$-Frank-Wolfe) use strongly-convex regularizers. Second, to also include CCCP [75] as a special case, we relax their Lipschitz smoothness assumption on $f$ to semi-concavity (which is weaker, as any $L$-smooth function is also $L$-concave). Third, we also consider much weaker stepsize schemes such as *constant* or *non-summable* ones. We show that for either concave $f$ or strongly-convex $g$, a better $\mathcal{O}(1/k)$ rate of convergence can be achieved, even under the (weak) constant stepsize. It should be noted that our results are new.

All the results in this section are stated under the following assumptions, where $L_f$ and $\sigma_g$ are non-negative constants and $\|\cdot\|$ denotes the $\ell_2$ norm. Their proofs are given in Appendix B.

**Assumption 1.** $f$ is differentiable and $L_f$-semi-concave (i.e., $f(\mathbf{x}) - \frac{L_f}{2}\|\mathbf{x}\|^2$ is concave) on $\mathrm{dom}\, f$, which is assumed to be open and convex. When $L_f = 0$, $f$ is concave.

**Assumption 2.** $g$ is proper, closed, and $\sigma_g$-strongly-convex (i.e., $g(\mathbf{x}) - \frac{\sigma_g}{2}\|\mathbf{x}\|^2$ is convex), and $\mathrm{dom}\, g \subseteq \mathrm{dom}\, f$ is compact. When $\sigma_g > 0$, $g$ is strongly convex.

Let $\mathbf{p_x}$ denote a solution of $\min_{\mathbf{p}}\{\langle \nabla f(\mathbf{x}), \mathbf{p}\rangle + g(\mathbf{p})\}$ and let $\mathbf{p}^k = \mathbf{p}_{\mathbf{x}^k}$. The following quantity, called the *conditional gradient norm* [7], will serve as an optimality measure:

$$S(\mathbf{x}) = \langle \nabla f(\mathbf{x}), \mathbf{x} - \mathbf{p_x}\rangle + g(\mathbf{x}) - g(\mathbf{p_x}). \tag{13}$$

**Lemma 1.** $S(\mathbf{x}) \geq \frac{\sigma_g}{2}\|\mathbf{x} - \mathbf{p_x}\|^2 \ \forall \mathbf{x} \in \mathrm{dom}\, f$, and $S(\mathbf{x}) = 0$ iff $\mathbf{x}$ is a stationary point of (9).

The following theorem contains our convergence results for the most common stepsize schemes, including the following *adaptive* and *line-search* stepsizes, respectively:

$$\alpha_k = \min\left\{1, \frac{1}{L_f + \sigma_g}\left(\frac{S(\mathbf{x}^k)}{\|\mathbf{p}^k - \mathbf{x}^k\|^2} + \frac{\sigma_g}{2}\right)\right\}, \qquad \alpha_k = \underset{\alpha \in [0,1]}{\mathrm{argmin}}\, F(\mathbf{x}^k + \alpha(\mathbf{p}^k - \mathbf{x}^k)). \tag{14}$$

**Theorem 1.** *Let $F^*$ be the minimum value of $F$, $\Omega$ be the diameter of $\mathrm{dom}\, g$, $\Delta_k = F(\mathbf{x}^k) - F^*$, $\omega = \frac{\sigma_g}{L_f + \sigma_g}$, $\rho(\alpha) = \alpha \min\{1, 2 - \frac{\alpha}{\omega}\}$, $\eta(\alpha) = \frac{1}{2}[(L_f + \sigma_g)\alpha - \sigma_g]$, and $\mu = \sqrt{2L_f \Delta_0}$. For any $k \geq 0$, we have $\min_{0 \leq i \leq k} S(\mathbf{x}^i) \leq B_k$, where the bound $B_k$ is given as follows:*

| | constant stepsize $\alpha_k = \alpha > 0\ \forall k$ | constant step length $\alpha_k = \frac{\alpha}{\|\mathbf{p}^k - \mathbf{x}^k\|}\ \forall k$ | non-summable $\sum_{k=0}^{+\infty}\alpha_k = \infty$ | adaptive or line search (14) |
|---|---|---|---|---|
| convex $g$ | $\frac{\Delta_0}{\alpha(k+1)} + \frac{L_f\Omega^2\alpha}{2}$ | $\frac{\Delta_0\Omega}{\alpha(k+1)} + \frac{L_f\Omega\alpha}{2}$ | $\frac{\Delta_0 + \frac{L_f\Omega^2}{2}\sum_{i=0}^{k}\alpha_i^2}{\sum_{i=0}^{k}\alpha_i}$ | $\max\left(\frac{2\Delta_0}{k+1}, \frac{\mu\Omega}{\sqrt{k+1}}\right)$ |
| strongly convex $g$ | $\frac{\Delta_0}{\alpha(k+1)} + \eta(\alpha)\Omega^2\ \forall\alpha \geq 2\omega$ $\frac{\Delta_0}{\rho(\alpha)(k+1)}\ \forall\alpha < 2\omega$ | $\left(\frac{\Delta_0}{\alpha\sqrt{2\sigma_g}(k+1)} + \frac{(L_f+\sigma_g)\alpha}{2\sqrt{2\sigma_g}}\right)^2$ | $\frac{\Delta_{k(\omega)}}{\sum_{i=k(\omega)}^{k}\alpha_i}$ | $\frac{\Delta_0}{\omega(k+1)}$ |
| concave $f$ | $\frac{\Delta_0}{\alpha(k+1)}$ | $\frac{\Delta_0\Omega}{\alpha(k+1)}$ | $\frac{\Delta_0}{\sum_{i=0}^{k}\alpha_i}$ | $\frac{2\Delta_0}{k+1}$ |

*In the above, $k(\omega) = \min\{k : \alpha_i < 2\omega\ \forall i \geq k\}$, with further assumption that $\lim_{k\to\infty}\alpha_k = 0$ for (jointly) non-concave $f$ and non-summable $\alpha_k$. For the non-highlighted cases, we have $\lim_{k\to\infty}S(\mathbf{x}^k) = 0$ and any limit point of the sequence $(\mathbf{x}^k)_{k\geq 0}$ is a stationary point of (9).*

The table in Theorem 1 also provides rates of convergence for the algorithm. Prior to our work, the $\mathcal{O}(1/\sqrt{k})$ rate for the adaptive or line-search stepsizes (top-right cell of the table, due to Beck [7])

---

[6]Note that $g$ is still assumed to be convex, so that the subproblem (10) can be solved to global optimality.

was the best for *nonconvex* objectives.[7] We have improved this rate to $\mathcal{O}(1/k)$ when $f$ is concave or $g$ is strongly convex, even under weaker stepsize schemes. In particular, convergence is guaranteed for all considered stepsize schemes when $f$ is concave, for which the best bound is obtained when $\alpha_k = 1 \; \forall k$, which explains the default unit stepsize in CCCP [75] (see §3.4). Convergence is also guaranteed for the (diminishing) non-summable scheme (which includes common stepsizes such as $\alpha_k = 2/(k+2)$ or $\alpha_k = 1/\sqrt{k}$), but the rate depends on the rate of divergence of $\sum_{i=0}^{k} \alpha_i$. More detailed results and analyses can be found in Appendix B.

**Convergence for MAP inference**    For all the instantiations of regularized Frank-Wolfe presented in §3.3 and §3.4, it is easy to check that Assumptions 1 and 2 are satisfied. In addition, the regularizers in most of them (euclidean or entropic variants, including mean field) are strongly convex, thus we would expect a rate of convergence of at least $\mathcal{O}(1/k)$ in practice for these algorithms under the adaptive, line search, or (suitable) constant stepsizes. Note that the adaptive scheme requires to know $L_f$ and $\sigma_g$, which is possible in our case: a lower bound on $\sigma_g$ is $\lambda$ for both euclidean and entropic variants, while an upper bound on $L_f$ is $\|\mathbf{P}\|_2$ for the energy (6). In practice, however, these bounds could be too loose to yield good convergence.

**Convergent mean field**    It is well-known that parallel mean field may diverge [35]. Our Entropic Frank-Wolfe can be viewed as an improved variant of mean field that is globally convergent for different stepsize schemes, without resorting to a concave approximation as done by Krähenbühl and Koltun [35]. Our above analysis also provides an explanation for a known phenomenon [6]: *damped mean field* (corresponding to Entropic Frank-Wolfe with $\lambda = 1$ and $\alpha_k = \alpha < 1 \; \forall k$) is more likely (than mean field) to guarantee convergence when the energy is not concave.

## 4.2    Tightness of the relaxation

We have seen that regularized Frank-Wolfe (Algorithm 1) minimizes a modified continuous energy. It is thus reasonable to ask whether doing so also minimizes the original discrete energy (which is the main objective). In this section, we partially answer this question by providing some tightness guarantee for this regularized continuous relaxation. Our analysis is quite general and also includes several existing tightness results [10, 41, 61] as special cases. All proofs can be found in Appendix C.2.

The last step in Algorithm 1 consists in converting $\mathbf{x}$ to a discrete solution. We consider two such rounding schemes. The simplest one is perhaps **nearest rounding**, which assigns each node $i$ with the label $s_i \in \operatorname{argmax}_{t \in \mathcal{S}} x_{it}$. Intuitively, this sets $\mathbf{x}_i$ to the nearest vertex of the simplex $\left\{ \mathbf{x}_i \in \mathbb{R}_+^d : \mathbf{1}^\top \mathbf{x}_i = 1 \right\}$. The second scheme, called **BCD rounding** [41, 61], consists in minimizing $E(\mathbf{x})$ over $\mathbf{x}_i$ while keeping all $\mathbf{x}_j$ ($j \neq i$) fixed (i.e., block coordinate descent), which amounts to iteratively assigning each node $i$ with label $s_i \in \operatorname{argmin}_{s \in \mathcal{S}} \left\{ \theta_i(s) + \sum_{j \in \mathcal{N}_i} \sum_{t \in \mathcal{S}} \theta_{ij}(s,t) x_{jt} \right\}$.
In practice, we only use nearest rounding because BCD rounding is too expensive for dense graphs. However, an important property of the latter is that it does not increase the energy, which is useful for our theoretical analysis. The following theorem provides an additive bound on the energy.

**Theorem 2.** *Let $\mathbf{x}_r^*$ be a global minimum of $E_r(\mathbf{x}) = E(\mathbf{x}) + r(\mathbf{x})$ over $\mathcal{X}$, $\bar{\mathbf{x}}_r^*$ be the discrete solution rounded from $\mathbf{x}_r^*$, and $E^*$ be the minimum discrete energy. Assume that $r(\mathbf{x})$ is bounded:[8] $m \leq r(\mathbf{x}) \leq M \; \forall \mathbf{x} \in \mathcal{X}$. We have $E^* \leq E(\bar{\mathbf{x}}_r^*) \leq E^* + M - m + C$, where $C = \sqrt{n \left(1 - \frac{1}{d}\right)} \left( \|\mathbf{u}\|_2 + \sqrt{n} \|\mathbf{P}\|_2 \right)$ for nearest rounding and $C = 0$ for BCD rounding.*

Let us derive the energy BCD bound for some particular cases (see §C.2 for details). Obviously with no regularization ($r = 0$), we have $M = m = 0$ and thus $E(\bar{\mathbf{x}}_r^*) \leq E^* \leq E(\bar{\mathbf{x}}_r^*)$, yielding $E(\bar{\mathbf{x}}_r^*) = E^*$, i.e., the relaxation is tight. We have thus recovered a previously known result [10, 41, 61]. For $r(\mathbf{x}) = -\mathbf{c}^\top \mathbf{x} + \mathbf{x}^\top \operatorname{diag}(\mathbf{c})\mathbf{x}$ with $\mathbf{c} \geq \mathbf{0}$, we have $M = 0$ and $m = -\frac{1}{4}\mathbf{1}^\top \mathbf{c}$, which recovers exactly the additive bound given by Ravikumar and Lafferty [61] for the convex QP relaxation. For the $\ell_2$ regularizer $r(\mathbf{x}) = \frac{\lambda}{2} \|\mathbf{x}\|_2^2$, we have $M = \frac{\lambda n}{2}$ and $m = \frac{\lambda n}{2d}$, thus we obtain a bound of $\frac{\lambda n}{2} \left(1 - \frac{1}{d}\right)$. For the entropy regularizer $r(\mathbf{x}) = -\lambda H(\mathbf{x})$, we have $M = 0$ and $m = -\lambda n \log d$, thus the bound is $\lambda n \log d$, which is worse than the $\ell_2$ bound for any $d \geq 5$.

---

[7]If both $f$ and $g$ are *convex*, a better rate of $\mathcal{O}(1/k)$ exists [5, 7, 74]. In addition, if $g$ is the indicator function of a convex set $\mathcal{X}$ (i.e., vanilla Frank-Wolfe) and either $f$ or $\mathcal{X}$ is strongly convex, a linear rate can be obtained [32, 50, 59]. Note that we use quite different machinery from all these analyses, due to the nonconvexity.

[8]A sufficient condition is $r$ being continuous, as $\mathcal{X}$ is compact.

Note that the bound provided by Theorem 2 is achieved from a *global* minimum of the regularized relaxation. This can be attained in some cases, e.g., when the energy is submodular or when the (convex) regularizer is large enough to make the objective convex. In the general case, however, the algorithm is only guaranteed to reach a stationary point, and the (theoretical) quality of such point remains unknown. It would be interesting to investigate whether the algorithm can provide an approximation guarantee for some classes of energies (e.g., supermodular ones), similar to some existing algorithms [11]. These open questions are left for future work.

## 5 Experiments

We compare regularized Frank-Wolfe with existing methods on the semantic segmentation task, in terms of both *inference* and *learning* performance. Two variants, namely Euclidean Frank-Wolfe ($\ell_2$**FW**) and Entropic Frank-Wolfe ($e$**FW**) (§3.3), will be compared to the following methods: Mean field (**MF**) [34, 35] (which is our baseline), nonconvex vanilla Frank-Wolfe (**FW**) [41] (§2.3), projected gradient descent (**PGD**) [39, 41], fast proximal gradient method (**PGM**) [9, 48], and alternating direction method of multipliers (**ADMM**) [40, 41]. Convex vanilla Frank-Wolfe [21] and (entropic) mirror descent [8, 55] were found to perform poorly in our experiments, and thus excluded from the presentation. Other methods based on CCCP [21] or LP relaxation [1, 21] are also excluded due to their sophisticated implementations. For all methods, we set the initial solution to $\mathbf{x}^0 = \text{softmax}(-\mathbf{u})$, following previous work [34]. Further details on implementation, running time, and memory footprint can be found in Appendices D–E.

### 5.1 Experimental setup

Our segmentation model is a standard combination of a CNN and a CRF [35, 76] (Appendix E.1). For the CNN part, we consider two strong architectures: DeepLabv3 with ResNet101 backbone [16], and DeepLabv3+ with Xception65 backbone [17]. The CRF part is a fully-connected one [34] in which any pair of pixels $(i, j)$ is an edge with potential $\theta_{ij}(s, t) = \mu(s, t)k(\mathbf{f}_i, \mathbf{f}_j) \, \forall s, t \in \mathcal{S}$, where $\mu : \mathcal{S} \times \mathcal{S} \to \mathbb{R}$ is called label compatibility function, and $k$ is a Gaussian kernel over image features based on pixel coordinates and colors. The setup of our models are similar to Zheng et al. [76]. We use the Potts compatibility function: $\mu(s, t) = w\mathbb{1}_{[s \neq t]}$ with $w = 1$ for the inference experiments in §5.2, and also for CRF initialization in the learning experiments in §5.3. For all experiments, a fully-trained CNN is needed. We follow closely the published recipes [16, 17] for this task. We first pretrain DeepLabv3 and DeepLabv3+ on the COCO dataset [46] and then finetune them on PASCAL VOC (*trainaug*) and Cityscapes (*train*) to obtain similar results to previous work [16, 17] (Table 1, CNN column). Finally, we perform experiments on two popular datasets: (augmented) PASCAL VOC [22] and Cityscapes [19]. Further details are given in Appendix E.

### 5.2 Inference performance

In this section, we compare the performance of regularized Frank-Wolfe against the competing methods in terms of inference. We consider a Potts CRF on top of a CNN, which is the typical setup for using dense CRF in post-processing. Figure 1a shows the *discrete* energy per inference iteration for each method, averaged over the 1449 *val* images of PASCAL VOC, using DeepLabv3+. One can observe that Frank-Wolfe variants completely outperform the other methods. In addition, regularized Frank-Wolfe outperforms all the other methods for a large range of $\lambda$, as shown in Figure 1b.

Table 1 shows the performance on the validation sets of PASCAL VOC and Cityscapes, for a Potts CRF with both DeepLabv3 and DeepLabv3+ as backbone. In this experiment, we run all the methods for 10 iterations. One can observe that $\ell_2$FW achieved the best performance, followed by $e$FW.

We should note some inconsistency compared to the energy results previously presented in Figure 1a. For example, $e$FW achieved much lower energy than MF, yet the mIoU gap is marginal; also, FW accuracy is slightly worse than MF while the energy is much better (lower). This can

|   |      | CNN   | PGD   | PGM   | ADMM  | MF    | FW    | $e$FW$_7$ | $e$FW$_3$ | $\ell_2$FW |
|---|------|-------|-------|-------|-------|-------|-------|-------|-------|-------|
| VOC | DL3  | 81.83 | 82.23 | 82.23 | 82.22 | 82.21 | 82.27 | 82.26 | **82.29** | **82.29** |
| VOC | DL3+ | 82.89 | 83.36 | 83.37 | 83.38 | 83.45 | 83.43 | 83.45 | 83.48 | **83.50** |
| CITY | DL3  | 76.73 | 76.88 | 76.86 | 76.95 | 76.97 | 76.86 | 76.99 | 76.99 | **77.03** |
| CITY | DL3+ | 79.55 | 79.64 | 79.63 | **79.66** | 79.63 | 79.64 | 79.65 | **79.66** | 79.66 |

Table 1: **Validation mIoU using a Potts CRF** on top of the pretrained CNN models. DL means DeepLab.

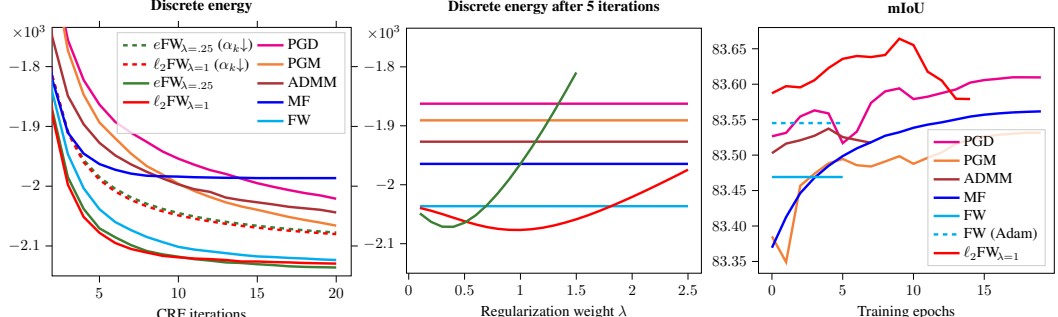

**(a)** Discrete energy comparison. **(b)** Effect of regularization weight. **(c)** Learning failure of vanilla FW.

**Figure 1: Results on PASCAL VOC validation set** using DeepLabv3+ and Potts dense CRF. **(a)** Comparison between CRF solvers ($\alpha_k\downarrow$ means $\alpha_k = k/(k+2)\ \forall k$) shows that Frank-Wolfe variants clearly outperform the other methods in terms of energy minimization. **(b)** Performance of regularized Frank-Wolfe can be greatly affected by $\lambda$, but it can still achieve lower energies than the other methods for a large range of $\lambda$. **(c)** Vanilla Frank-Wolfe completely fails to learn because of the zero-gradient issue.

be explained by the fact that the Potts model is not a perfect representation (i.e., lower energy in this model does not necessarily translate to higher accuracy). In the next section, we will see how the methods perform when the CRF parameters are learned from data.

### 5.3 Learning performance

In this section, we evaluate the performance of the methods for joint CNN-CRF end-to-end training. The CNN is initialized with its fully-trained weights on the corresponding dataset, and the CRF is initialized with the Potts model with random noise added. We train the model for 20 epochs with 5 CRF iterations,[9] using the same poly schedule as before. As the CNN has been already fully-trained, we set its learning rate to a small value of $0.0001$. For the CRF, we tried 4 different values of initial learning rates $\eta_0 \in \{1.0, 0.1, 0.01, 0.001\}$ and found that $1.0$ is too high (training diverges quickly) while $0.001$ is too low (slow progress) for all methods. For the remaining candidates $\{0.1, 0.01\}$, we perform 4 additional trainings for each method (i.e., a total of 5 runs for each configuration).

Let us summarize our findings. First, we observe that (vanilla) FW fails to learn. This is illustrated in Figure 1c, where we show the validation accuracy per epoch on PASCAL VOC for each method: FW did not make any progress. We tried a different optimizer (Adam [33]) and obtained similar results. This is expected as the gradient in vanilla FW is zero almost everywhere, as previously discussed in §3.1 (see also §C.1). Our second observation is that training is quite unstable for PGD, PGM, ADMM, $e\mathrm{FW}_{.3}$, and $\ell_2\mathrm{FW}$. In particular, $\eta_0 = 0.1$ is still too high for these methods, and even with $\eta_0 = 0.01$, some of the runs produced bad results. By contrast, MF and $e\mathrm{FW}_{.7}$ are stable for both learning rates, with $0.1$ being slightly better. A possible explanation is that PGD, PGM, ADMM, and $\ell_2\mathrm{FW}$ all employ a simplex projection step that is not fully differentiable (but only so almost everywhere). For $e\mathrm{FW}_{.3}$ (which is fully differentiable), we hypothesize that the low regularization makes the problem less "smooth", which may also harm gradient-based training. Finally, with the above training scheme, we observe that none of the CRF methods could improve over the CNN (but rather the opposite) on Cityscapes. We have seen that the Potts CRF was able to achieve some marginal improvements (Table 1), thus it is reasonable to expect even better performance with end-to-end training.

In view of the above observations, we present a simple trick to make CRF training more stable. The idea is to replace the CRF output $\mathbf{x}^*$ with $\frac{1}{2}(\mathbf{x}^* + \mathbf{x}^0)$, where we recall that the initialization $\mathbf{x}^0$ is the $\mathrm{softmax}$ of the CNN logits. Intuitively, this adds a skip connection from the CNN to the CRF output in the computation graph, which makes the gradient of the loss propagate directly to the CNN. We found that this trick also slightly improves $e\mathrm{FW}_{.7}$, but has a negative effect on MF. Therefore, it is applied to all methods except MF. Finally, as Cityscapes requires a very high number of epochs, we set this value to 100. Also because training on Cityscapes requires a lot more computing resources, we only perform a single run on DeepLabv3+. The results are presented in Table 2.

---

[9]While we use the same number of iterations at *test time* to simplify the evaluation protocol, it should be noted that using more iterations could be beneficial. See Appendix F.3 for some results.

Again, $\ell_2$FW consistently achieves the best results. Interestingly, while $e$FW$_3$ achieved similar performance to $\ell_2$FW in terms of energy minimization (Figure 1a and Table 1), its performance is worse in joint training. Compared to Table 1, we see that joint training produced much larger improvements over the CNNs, up to $2.25\%$ on PASCAL VOC and $0.4\%$ on Cityscapes.

|  |  | CNN | PGD | PGM | ADMM | MF | $e$FW$_7$ | $e$FW$_3$ | $\ell_2$FW |
|---|---|---|---|---|---|---|---|---|---|
| VOC | DL3 | 81.83 | 83.69 ±0.20 | **83.75** ±0.23 | 83.68 ±0.06 | 83.69 ±0.10 | 83.50 ±0.10 | 83.25 ±0.20 | **83.75** ±0.13 |
|  | DL3+ | 82.89 | 84.82 ±0.23 | 84.79 ±0.20 | 84.83 ±0.06 | 84.87 ±0.17 | 84.64 ±0.23 | 84.50 ±0.16 | **85.14** ±0.09 |
| CITY | DL3+ | 79.55 | 79.80 | 79.62 | 79.62 | 79.74 | 79.70 | 79.58 | **79.95** |

**Table 2: Validation mIoU under joint training**. For PASCAL VOC, we report the mean and standard deviation from 5 runs.

**Performance on the test sets**  We select the best performing method (DeepLabv3+ with $\ell_2$FW CRF) for evaluation on the test sets. For PASCAL VOC, we further train our model on the union of the *train* and *val* subsets for 50 epochs. For Cityscapes, we further train 200 epochs on *train* and *train_extra*, using the high-quality annotations provided by Tao et al. [70] (for *train_extra*). At the $150^{\text{th}}$ epoch, we replace *train_extra* with *val*. For this fine-tuning step, learning rates were set to 0.001 for CNN and 0.1 for CRF. For prediction, we apply test time augmentation including left-right flipping and multi-scales. For reference, we train DeepLabv3+ alone using the same recipes. Table 3 shows that we were able to closely match the performance reported by Chen et al. [17]. Adding the $\ell_2$FW CRF yields an improvement of 0.4 points on PASCAL VOC. Unfortunately we only observe a marginal improvement on Cityscapes.

| Model | VOC | CITY |
|---|---|---|
| DeepLabv3+ [17] | 87.8 | 82.1 |
| DeepLabv3+ (this work) | 87.6 | 83.5 |
| DeepLabv3+ with $\ell_2$FW CRF | **88.0** | **83.6** |

**Table 3:** Performance on the *test* sets. Submission URLs are given in Appendix F.1.

### 5.4 Ablation studies

**Trainable $\alpha_k$ and $\lambda$**  It is possible to learn $\alpha_k$ and $\lambda$ from data by simply setting them to be *trainable*. We carried out such an experiment with $\ell_2$FW and $e$FW but did not observe significant improvements, though we should note that a more sophisticated training recipe (e.g., using custom learning rates for these variables) might lead to better results. Details are provided in Appendix F.2.

**Fine-grained analysis**  We observe that CRF improved over CNN on most of the semantic classes. In particular, on bicycle (known to be the most challenging class of PASCAL VOC [16]), $\ell_2$FW and $e$FW achieved improvements of over $10\%$ absolute in mIoU. See Appendix F.3 for the details.

## 6 Discussion & conclusion

**Why does it work?**  Theoretically, all the methods in §5 should reach a stationary point, so how can one be better than another? In fact, Figure 1a only shows that Frank-Wolfe variants work better than the other methods *in the first few iterations*, but not necessarily in a later stage. Indeed, the same conclusion no longer holds after 100 iterations (see §F.4), but this long regime is not practical because it would lead to vanishing/exploding gradients [76] and to potentially prohibitive memory consumption. As to why Frank-Wolfe achieves lower energy in the early stage, we hypothesize that this could be due to the discreteness of its iterates (7). With small $\lambda$, the solution by regularized Frank-Wolfe should be close to the vanilla one, and thus also benefits from this property. It is important to note that the benefit of regularized Frank-Wolfe does not lie in the extra (sometimes small) energy improvement over vanilla Frank-Wolfe, but in its ability to seamlessly solve the zero-gradient issue.

**How to tune $\lambda$?**  We found that similar curves to Figure 1b can be obtained using a small random subset (e.g., 10 samples) of the data, which suggests a quick way of tuning $\lambda$ by random subsampling. In practice, this step takes only a few seconds, which is negligible in most training scenarios.

**Limitations**  While one variant of regularized Frank-Wolfe ($\ell_2$FW) consistently achieves the best results, the difference compared to the other methods is sometimes small. In addition, the improvement of dense CRFs over CNNs is marginal on the Cityscapes test set. Nevertheless, we hope the encouraging results on PASCAL VOC could attract interest from the community in CRF research, potentially leading to creative ways of overcoming these limitations.

**Societal impact**  Semantic segmentation models can be used in surveillance systems, which might raise potential privacy concerns. Furthermore, the datasets that our models were trained on are known to present strong built-in bias [71], thus they should be used with caution.

## Acknowledgments and Disclosure of Funding

This work was supported in part by the ANR grant AVENUE (ANR-18-CE23-0011), and was partly done when the first author was affiliated with Manifold Perception (`mption.com`). The experiments were performed using HPC resources from GENCI-IDRIS (Grants 2020-AD011011321 and 2020-AD011011881). The authors thank the anonymous reviewers and meta-reviewer for their constructive feedback that helped improve the manuscript.

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
