# Appendices

## A   Details on special cases of regularized Frank-Wolfe inference

We have seen in §3 multiple instantiations of regularized Frank-Wolfe, leading to new algorithms for MAP inference, as well as recovering many existing ones. In this section we provide further details on this matter.

Recall the notation $n = |\mathcal{V}|, m = |\mathcal{E}|, d = |\mathcal{S}|$, where $\mathcal{V}, \mathcal{E}$ and $\mathcal{S}$ are the sets of nodes, edges, and labels, respectively.

### A.1   Algorithms based on QP relaxation with vanilla Frank-Wolfe

**Nonconvex vanilla Frank-Wolfe**   This algorithm, previously studied by Lê-Huu and Paragios [41], was already presented in §2.3. It consists in applying vanilla Frank-Wolfe directly to the energy (6).

**Convex vanilla Frank-Wolfe**   This involves the *convex* QP relaxation of MAP inference introduced by Ravikumar and Lafferty [61]. The idea is to add a sufficiently large vector $\mathbf{c}$ to the diagonal of $\mathbf{P}$ to make it positive semidefinite. If $\mathbf{x} \in \{0,1\}^{nd}$ then it is easy to check that $\mathbf{x}^\top \operatorname{diag}(\mathbf{c})\mathbf{x} = \mathbf{c}^\top \mathbf{x}$ for any $\mathbf{c} \in \mathbb{R}^{nd}$. Therefore, the (discrete) energy can be written as

$$E(\mathbf{x}) = \frac{1}{2}\mathbf{x}^\top (2\operatorname{diag}(\mathbf{c}) + \mathbf{P})\mathbf{x} + (\mathbf{u} - \mathbf{c})^\top \mathbf{x}. \tag{15}$$

It can be shown that the above function is convex if $\mathbf{c}$ is chosen as follows:

$$c_{is} = \frac{1}{2}\sum_{j \in \mathcal{N}_i}\sum_{t \in \mathcal{S}}\theta_{ij}(s,t) \quad \forall i \in \mathcal{V}, s \in \mathcal{S}, \tag{16}$$

where $\mathcal{N}_i$ denotes the set of neighbors of node $i$. Applying vanilla Frank-Wolfe to minimizing the above convex energy over $\mathcal{X}$ yields the algorithm presented in section 4 of Desmaison et al. [21].

## A.2 Algorithms based on LP relaxation with vanilla Frank-Wolfe

Let us first present the LP relaxation of MAP inference. We use the same notation leading to the energy formulation (4), namely the indicator variables $x_{is} \in \{0, 1\}$, the indicator vectors $\mathbf{x}_i \in \{0, 1\}^d$, and the potential vectors $\boldsymbol{\theta}_i \in \mathbb{R}^d$ for all nodes $i \in \mathcal{V}$ and labels $s \in \mathcal{S}$. In addition, define for all edges $ij \in \mathcal{E}$ and pairs of labels $(s, t) \in \mathcal{S}^2$:

- New pairwise indicator variables $x_{ijst} = x_{is} x_{jt} \in \{0, 1\}$.

- New pairwise indicator vectors $\mathbf{x}_{ij} = (x_{ijst})_{s \in \mathcal{S}, t \in \mathcal{S}} \in \{0, 1\}^{d^2}$.

- New pairwise potential vectors $\boldsymbol{\theta}_{ij} = (\theta_{ij}(s, t))_{s \in \mathcal{S}, t \in \mathcal{S}} \in \mathbb{R}^{d^2}$, which can be viewed as the flatten version of the potential matrices $\boldsymbol{\Theta}_{ij}$ in (4).

Then, the energy (4) can be rewritten as a linear function:

$$E_{\mathrm{LP}}(\mathbf{x}; \boldsymbol{\theta}) = \sum_{i \in \mathcal{V}} \boldsymbol{\theta}_i^\top \mathbf{x}_i + \sum_{ij \in \mathcal{E}} \boldsymbol{\theta}_{ij}^\top \mathbf{x}_{ij}, \tag{17}$$

where by slight abuse of notation, we let $\mathbf{x}$ and $\boldsymbol{\theta}$ again denote the vectors of all variables and parameters, respectively. Note that $\mathbf{x}$ and $\boldsymbol{\theta}$ are now $(nd + md^2)$-dimensional vectors and not $nd$-dimensional as in (4). The LP relaxation consists in minimizing $E_{\mathrm{LP}}$ over the following *local polytope* [72]:

$$\mathcal{X}_{\mathrm{LP}} = \left\{ \mathbf{x} \in \mathbb{R}^{nd+md^2} \,\middle|\, \begin{array}{c} \mathbf{x} \geq \mathbf{0}, \\ \mathbf{1}^\top \mathbf{x}_i = 1 \; \forall i \in \mathcal{V}, \\ \displaystyle\sum_{t \in \mathcal{S}} x_{ijst} = x_{is} \; \forall ij \in \mathcal{E}, \forall s \in \mathcal{S}, \\ \displaystyle\sum_{s \in \mathcal{S}} x_{ijst} = x_{jt} \; \forall ij \in \mathcal{E}, \forall t \in \mathcal{S}. \end{array} \right\}. \tag{18}$$

The last two constraints in the above (called *local consistency*) can be written as $\mathbf{A}\mathbf{x} = \mathbf{0}$ for some $(2md) \times (nd + md^2)$ matrix $\mathbf{A}$. We can thus rewrite the LP relaxation compactly as:

$$\min E_{\mathrm{LP}}(\mathbf{x}; \boldsymbol{\theta}) \triangleq \boldsymbol{\theta}^\top \mathbf{x}, \quad \text{s.t.} \quad \mathbf{x} \in \mathcal{X}_{\mathrm{LP}} \triangleq \left\{ \mathbf{x} \in \mathbb{R}_+^{nd+md^2} : \mathbf{1}^\top \mathbf{x}_i = 1 \; \forall i \in \mathcal{V}, \mathbf{A}\mathbf{x} = \mathbf{0} \right\}. \tag{19}$$

As presented in §3.4, Sontag and Jaakkola, Meshi et al., Tang et al. [67, 52, 69] apply vanilla Frank-Wolfe to minimize a regularized LP energy:

$$\min_{\mathbf{x} \in \mathcal{X}_{\mathrm{LP}}} E_{\mathrm{LP}}(\mathbf{x}; \boldsymbol{\theta}) + r(\mathbf{x}) \tag{20}$$

for some regularizer $r$. These works differ in the choice of $r$.

**Local-consistency regularization** Choosing $r(\mathbf{x}) = \frac{\lambda}{2} \|\mathbf{A}\mathbf{x}\|_2^2$ we obtain the algorithm presented by Meshi et al. [52] (which corresponds to the primal algorithm in the top-right cell of their Table 1).

**Bethe and TRW entropic regularization** Sontag and Jaakkola [67] also apply vanilla Frank-Wolfe to a regularized LP energy (corresponding to Step 3 in their Algorithm 1; note that we consider only the first outer iteration of their algorithm). They consider regularizers of the form

$$r(\mathbf{x}) = -\tilde{H}(\mathbf{x}), \tag{21}$$

where $\tilde{H}(\mathbf{x})$ is some approximation to the entropy $H(\mathbf{x})$ of the distribution over $\mathbf{x}$.

Define the singleton entropy

$$H(\mathbf{x}_i) = -\sum_{s \in \mathcal{S}} x_{is} \log x_{is} \quad \forall i \in \mathcal{V}, \tag{22}$$

and the pairwise mutual information

$$I(\mathbf{x}_{ij}) = \sum_{s \in \mathcal{S}} \sum_{t \in \mathcal{S}} x_{ijst} \log \frac{x_{ijst}}{x_{is} x_{jt}} = -H(\mathbf{x}_{ij}) + H(\mathbf{x}_i) + H(\mathbf{x}_j) \quad \forall ij \in \mathcal{E}. \tag{23}$$

The so-called Bethe approximation is defined as:

$$\tilde{H}_{\text{Bethe}}(\mathbf{x}) = \sum_{i \in \mathcal{V}} H(\mathbf{x}_i) - \sum_{ij \in \mathcal{E}} I(\mathbf{x}_{ij}). \tag{24}$$

The second approximation considered by [67] is called tree-reweighted (TRW) approximation. To achieve this, we decompose the the graph into a convex combination of spanning trees according to some distribution (over the trees), and let $\rho_{ij}$ be the so-called *edge appearance probability*, which is computed as the number of spanning trees containing the edge $ij$ in the current decomposition, divided by the total number of all possible spanning trees containing $ij$ (in the entire distribution). The TRW approximation is then given by

$$\tilde{H}_{\text{TRW}}(\mathbf{x}) = \sum_{i \in \mathcal{V}} H(\mathbf{x}_i) - \sum_{ij \in \mathcal{E}} \rho_{ij} I(\mathbf{x}_{ij}). \tag{25}$$

**$\rho$-reweighted entropic regularization**  Tang et al. [69] consider a more general term than the previous ones, based on the following approximation to $z \log z$ for $z \in [0, 1]$, parameterized by $\eta \in [0, 1]$:

$$H_\eta(z) = \begin{cases} -z \log z & \text{if } z \in [\eta, 1], \\ -\eta \log \eta - (1 + \log \eta)(z - \eta) - \frac{(z-\eta)^2}{2\eta} & \text{if } z \in [0, \eta]. \end{cases} \tag{26}$$

Define a similar version for vectors:

$$H_\eta(\mathbf{z}) = \sum_{i=1}^{p} H_\eta(z_i) \quad \forall \mathbf{z} \in \mathbb{R}^p. \tag{27}$$

Their **$\rho$-reweighted** approximation to the entropy $H(\mathbf{x})$ is given by:

$$\tilde{H}_\eta^\rho(\mathbf{x}) = \sum_{i \in \mathcal{V}} H_\eta(\mathbf{x}_i) - \sum_{ij \in \mathcal{E}} \rho_{ij} \left[ -H_\eta(\mathbf{x}_{ij}) + H_\eta(\mathbf{x}_i) + H_\eta(\mathbf{x}_j) \right] \tag{28}$$

Tang et al. [69] apply vanilla Frank-Wolfe to $E_{\text{LP}} + r$ where $r = -\tilde{H}_\eta^\rho$. Note that their work consists in learning parameters of graphical models through maximum likelihood estimation. Here we only consider the inference part presented in their Section 3.2, which is used as a subroutine for learning.

### A.3  Algorithms based on the concave-convex procedure

In the dense CRF model proposed by Krähenbühl and Koltun [34], the pairwise potentials consist of weighted sums of Gaussian kernels:

$$\theta_{ij}(s, t) = \sum_{c=1}^{C} \mu^{(c)}(s, t) k^{(c)}(\mathbf{f}_i, \mathbf{f}_j) \quad \forall i, j \in \mathcal{V}, \forall s, t \in \mathcal{S}, \tag{29}$$

where $C$ is the number of components, $\mu^{(c)} : \mathcal{S} \times \mathcal{S} \to \mathbb{R}$ are the so-called label compatibility functions, and $k^{(c)}$ are Gaussian kernels over some image features $(\mathbf{f}_i, \mathbf{f}_j)$ (§E.1 presents a concrete example implemented for our experiments).

Define kernel matrices $\mathbf{K}^{(c)} \in \mathbb{R}^{n \times n}$ with elements $K_{ij}^{(c)} = k(\mathbf{f}_i, \mathbf{f}_j)$ and compatibility matrices $\mathbf{M}^{(c)} \in \mathbb{R}^{d \times d}$ with elements $M_{st}^{(c)} = \mu^{(c)}(s, t)$. Let

$$\mathbf{M} = \sum_{c=1}^{C} \mathbf{M}^{(c)}. \tag{30}$$

If we assume that $\mathbf{K}^{(c)} \in \mathbb{R}^{n \times n}$ has unit diagonal: $K_{ii}^{(c)} = 1 \ \forall i, \forall c$, then our pairwise potential matrix $\mathbf{P}$ can be written as

$$\mathbf{P} = \sum_{c=1}^{C} \left( \mathbf{K}^{(c)} - \mathbf{I}_n \right) \otimes \mathbf{M}^{(c)} = -\mathbf{I}_n \otimes \mathbf{M} + \sum_{c=1}^{C} \mathbf{K}^{(c)} \otimes \mathbf{M}^{(c)}, \tag{31}$$

where $\otimes$ denotes the Kronecker product, and $\mathbf{I}_n$ is the $n \times n$ identity matrix.

In the following, the concave-convex procedure (CCCP) [75] is applied to minimizing $f(\mathbf{x}) + g(\mathbf{x})$ where $f$ is concave and $g$ is convex. (We integrate the constraint set $\mathcal{X}$ into $g$ using its indicator function $\delta_{\mathcal{X}}$, for consistency with our presentation of regularized Frank-Wolfe.)

**Convergent mean field**  Krähenbühl and Koltun [35] proposed (in their section 3.1) to minimizing a regularized energy $E(\mathbf{x}) + \mathbf{x}^\top \log \mathbf{x}$ (entropic regularizer) by applying CCCP to:

$$f(\mathbf{x}) = \frac{1}{2}\mathbf{x}^\top (\mathbf{P} + \mathbf{I}_n \otimes \mathbf{M})\mathbf{x} + \mathbf{u}^\top \mathbf{x}, \tag{32}$$

$$g(\mathbf{x}) = -\frac{1}{2}\mathbf{x}^\top (\mathbf{I}_n \otimes \mathbf{M})\mathbf{x} + \mathbf{x}^\top \log \mathbf{x} + \delta_{\mathcal{X}}(\mathbf{x}). \tag{33}$$

**Convergent mean field using concave approximation**  Krähenbühl and Koltun [35] proposed (in their section 3.2) a more efficient algorithm using:

$$f(\mathbf{x}) = \frac{1}{2}\mathbf{x}^\top (\mathbf{P} + \mathbf{I}_n \otimes \mathbf{M})\mathbf{x} + \mathbf{u}^\top \mathbf{x}, \tag{34}$$

$$g(\mathbf{x}) = \mathbf{x}^\top \log \mathbf{x} + \delta_{\mathcal{X}}(\mathbf{x}) \tag{35}$$

**CCCP for QP relaxation 1**  Desmaison et al. [21] proposed (in their section 5.1) the following application of CCCP:

$$f(\mathbf{x}) = -\mathbf{x}^\top \operatorname{diag}(c)\mathbf{x}, \tag{36}$$

$$g(\mathbf{x}) = \frac{1}{2}\mathbf{x}^\top (2\operatorname{diag}(\mathbf{c}) + \mathbf{P})\mathbf{x} + \mathbf{u}^\top \mathbf{x} + \delta_{\mathcal{X}}(\mathbf{x}), \tag{37}$$

where $\mathbf{c}$ is defined by (16).

**CCCP for QP relaxation 2**  Inspired by Krähenbühl and Koltun [35], Desmaison et al. [21] also proposed (in their section 5.2) another more efficient variant:

$$f(\mathbf{x}) = \frac{1}{2}\mathbf{x}^\top (\mathbf{P} + \mathbf{I}_n \otimes \mathbf{M})\mathbf{x}, \tag{38}$$

$$g(\mathbf{x}) = -\frac{1}{2}\mathbf{x}^\top (\mathbf{I}_n \otimes \mathbf{M})\mathbf{x} + \mathbf{u}^\top \mathbf{x} + \delta_{\mathcal{X}}(\mathbf{x}). \tag{39}$$

### A.4  Summary of special cases

We provide in Table 4 a summary of special cases discussed in this section as well as in §3.3 and §3.4. There we show how these algorithms can be obtained from regularized Frank-Wolfe by suitably choosing $f, g$ and $r$ in Algorithm 1. Recall that $f + g = E + r + \delta_{\mathcal{X}}$.

## B  Detailed convergence analysis

In this section, let $\|\cdot\|$ denote the $\ell_2$ norm. The following lemma is useful for the proofs.

**Lemma 2.** *If $f$ and $g$ satisfy Assumption 1 and 2, then*

$$f(\mathbf{y}) \leq f(\mathbf{x}) + \langle \nabla f(\mathbf{x}), \mathbf{y} - \mathbf{x} \rangle + \frac{L_f}{2}\|\mathbf{y} - \mathbf{x}\|^2 \quad \forall \mathbf{x}, \mathbf{y} \in \operatorname{dom} f, \tag{40}$$

$$g(\mathbf{y}) \geq g(\mathbf{x}) + \langle \mathbf{d}, \mathbf{y} - \mathbf{x} \rangle + \frac{\sigma_g}{2}\|\mathbf{y} - \mathbf{x}\|^2 \quad \forall \mathbf{x}, \mathbf{y} \in \operatorname{dom} g, \forall \mathbf{d} \in \partial g(\mathbf{x}). \tag{41}$$

*Proof.* For a convex function $h$, we have

$$h(\mathbf{y}) \geq h(\mathbf{x}) + \langle \mathbf{d}, \mathbf{y} - \mathbf{x} \rangle \quad \forall \mathbf{x}, \mathbf{y}, \forall \mathbf{d} \in \partial h(\mathbf{x}). \tag{42}$$

Applying the above inequality with, respectively, $h(\mathbf{x}) = -f(\mathbf{x}) + \frac{L_f}{2}\|\mathbf{x}\|_2^2$ and $h(\mathbf{x}) = g(\mathbf{x}) - \frac{\sigma_g}{2}\|\mathbf{x}\|_2^2$, we obtain (40) and (41). (Note that for the second case, $h$ is convex and thus $\partial g(\mathbf{x}) = \partial h(\mathbf{x}) + \sigma_g \mathbf{x}$.) $\qquad \square$

| Algorithm | $f(\mathbf{x})$ | $g(\mathbf{x}) - \delta_{\mathcal{X}}(\mathbf{x})$ |
|---|---|---|
| Parallel mean field Krähenbühl and Koltun [34] | $E(\mathbf{x})$ | $\mathbf{x}^\top \log \mathbf{x}$ |
| Convergent mean field 1 §3.1 in Krähenbühl and Koltun [35] | $E(\mathbf{x}) + \frac{1}{2}\mathbf{x}^\top (\mathbf{I}_n \otimes \mathbf{M}) \mathbf{x}$ | $-\frac{1}{2}\mathbf{x}^\top (\mathbf{I}_n \otimes \mathbf{M}) \mathbf{x} + \mathbf{x}^\top \log \mathbf{x}$ |
| Convergent mean field 2 §3.2 in Krähenbühl and Koltun [35] | $E(\mathbf{x}) + \frac{1}{2}\mathbf{x}^\top (\mathbf{I}_n \otimes \mathbf{M}) \mathbf{x}$ | $\mathbf{x}^\top \log \mathbf{x}$ |
| Nonconvex Frank-Wolfe Lê-Huu and Paragios [41] | $E(\mathbf{x})$ | $0$ |
| Convex Frank-Wolfe §4 in Desmaison et al. [21] | $E(\mathbf{x}) - \mathbf{c}^\top \mathbf{x} + \mathbf{x}^\top \operatorname{diag}(\mathbf{c})\mathbf{x}$ | $0$ |
| CCCP for QP relaxation 1 §5.1 in Desmaison et al. [21] | $-\mathbf{x}^\top \operatorname{diag}(\mathbf{c})\mathbf{x}$ | $E(\mathbf{x}) + \mathbf{x}^\top \operatorname{diag}(\mathbf{c})\mathbf{x}$ |
| CCCP for QP relaxation 2 §5.2 in Desmaison et al. [21] | $E(\mathbf{x}) + \frac{1}{2}\mathbf{x}^\top (\mathbf{I}_n \otimes \mathbf{M}) \mathbf{x} - \mathbf{u}^\top \mathbf{x}$ | $-\frac{1}{2}\mathbf{x}^\top (\mathbf{I}_n \otimes \mathbf{M})\mathbf{x} + \mathbf{u}^\top \mathbf{x}$ |
| LP local-consistency Frank-Wolfe Meshi et al. [52] | $E_{\mathrm{LP}}(\mathbf{x}) + \frac{\lambda}{2} \|\mathbf{A}\mathbf{x}\|_2^2$ | $0$ |
| LP Bethe Frank-Wolfe Sontag and Jaakkola [67] | $E_{\mathrm{LP}}(\mathbf{x}) + \tilde{H}_{\mathrm{Bethe}}(\mathbf{x})$ | $0$ |
| LP TRW Frank-Wolfe Sontag and Jaakkola [67] | $E_{\mathrm{LP}}(\mathbf{x}) + \tilde{H}_{\mathrm{TRW}}(\mathbf{x})$ | $0$ |
| LP $\boldsymbol{\rho}$-reweighted Frank-Wolfe Tang et al. [69] | $E_{\mathrm{LP}}(\mathbf{x}) + \tilde{H}_\eta^\rho(\mathbf{x})$ | $0$ |
| Euclidean Frank-Wolfe (This work) | $E(\mathbf{x})$ | $\frac{\lambda}{2} \|\mathbf{x}\|_2^2$ |
| Entropic Frank-Wolfe (This work) | $E(\mathbf{x})$ | $\lambda \mathbf{x}^\top \log \mathbf{x}$ |
| Lasso Frank-Wolfe (This work, not implemented) | $E(\mathbf{x})$ | $\lambda \|\mathbf{x}\|_1$ |

**Table 4:** Summary of special cases of regularized Frank-Wolfe.

## B.1 Proof of Lemma 1

First we show that

$$S(\mathbf{x}) \geq \frac{\sigma_g}{2} \|\mathbf{x} - \mathbf{p_x}\|^2 \ \forall \mathbf{x} \in \operatorname{dom} f. \tag{43}$$

Notice that

$$\mathbf{p_x} \in \underset{\mathbf{p}}{\operatorname{argmin}} \{\langle \nabla f(\mathbf{x}), \mathbf{p} \rangle + g(\mathbf{p})\} \iff -\nabla f(\mathbf{x}) \in \partial g(\mathbf{p_x}). \tag{44}$$

Hence, applying (41) we obtain

$$g(\mathbf{x}) \geq g(\mathbf{p_x}) + \langle -\nabla f(\mathbf{x}), \mathbf{x} - \mathbf{p_x} \rangle + \frac{\sigma_g}{2} \|\mathbf{x} - \mathbf{p_x}\|^2, \tag{45}$$

which is precisely (43).

To complete the proof, we need to show that $S(\mathbf{x}^*) = 0$ if and only if $\mathbf{x}^*$ is a stationary point of (9), i.e., $-\nabla f(\mathbf{x}^*) \in \partial g(\mathbf{x}^*)$. The following is due to Beck [7]. Notice that

$$S(\mathbf{x}) = \max_{\mathbf{p}} \{\langle \nabla f(\mathbf{x}), \mathbf{x} - \mathbf{p} \rangle + g(\mathbf{x}) - g(\mathbf{p})\}, \tag{46}$$

we have

$$S(\mathbf{x}^*) = 0 \iff S(\mathbf{x}^*) \leq 0 \iff \langle \nabla f(\mathbf{x}^*), \mathbf{x}^* - \mathbf{p} \rangle + g(\mathbf{x}^*) - g(\mathbf{p}) \leq 0 \ \forall \mathbf{p} \tag{47}$$

$$\iff g(\mathbf{p}) \geq g(\mathbf{x}^*) + \langle -\nabla f(\mathbf{x}^*), \mathbf{p} - \mathbf{x}^* \rangle \ \forall \mathbf{p} \tag{48}$$

$$\iff -\nabla f(\mathbf{x}^*) \in \partial g(\mathbf{x}^*). \tag{49}$$

The proof is completed.

## B.2   Proof of Theorem 1

We need an additional lemma.

**Lemma 3.** *For any* $\mathbf{x} \in \operatorname{dom} f$ *and any* $\alpha \in [0, 1]$ *we have*

$$F(\mathbf{x} + \alpha(\mathbf{p_x} - \mathbf{x})) - F(\mathbf{x}) \leq -\alpha S(\mathbf{x}) + K(\alpha) \|\mathbf{p_x} - \mathbf{x}\|^2, \tag{50}$$

*where* $K(\alpha) = \frac{1}{2} \left[ (L_f + \sigma_g)\alpha^2 - \sigma_g \alpha \right]$.

*Proof.* On one hand, from (40) we have

$$f(\mathbf{x} + \alpha(\mathbf{p_x} - \mathbf{x})) \leq f(\mathbf{x}) + \alpha \langle \nabla f(\mathbf{x}), \mathbf{p_x} - \mathbf{x} \rangle + \frac{L_f \alpha^2}{2} \|\mathbf{p_x} - \mathbf{x}\|^2. \tag{51}$$

On the other hand, from the $\sigma_g$-strong-convexity of $g$:

$$g(\mathbf{x} + \alpha(\mathbf{p_x} - \mathbf{x})) \leq (1 - \alpha)g(\mathbf{x}) + \alpha g(\mathbf{p_x}) - \frac{\sigma_g \alpha(1 - \alpha)}{2} \|\mathbf{p_x} - \mathbf{x}\|^2. \tag{52}$$

Summing up the above two inequalities, we obtain (50). $\square$

Let $S_k = S(\mathbf{x}^k)$, $r_k = \|\mathbf{p}^k - \mathbf{x}^k\|^2$, and $F_k = F(\mathbf{x}^k)$. Applying (50) we have

$$\boxed{F_k - F_{k+1} \geq \alpha_k S_k - K(\alpha_k) r_k.} \tag{53}$$

Therefore,

$$\Delta_0 = F_0 - F^* \geq F_0 - F_{k+1} = \sum_{i=0}^{k}(F_i - F_{i+1}) \geq S \sum_{i=0}^{k} \alpha_i - \sum_{i=0}^{k} r_i K(\alpha_i), \tag{54}$$

which implies

$$S \leq \frac{\Delta_0 + \sum_{i=0}^{k} r_i K(\alpha_i)}{\sum_{i=0}^{k} \alpha_i}. \tag{55}$$

This is an important inequality that will help us to obtain the convergence results for the weak stepsize schemes such as constant and non-summable ones.

For the adaptive (and line-search) stepsizes, the following observations will be useful. Notice that the RHS of (50) can be written as $\frac{1}{2}at^2 - bt$ where

$$a = (L_f + \sigma_g) \|\mathbf{p_x} - \mathbf{x}\|^2, \quad b = S(\mathbf{x}) + \frac{\sigma_g}{2} \|\mathbf{p_x} - \mathbf{x}\|^2.$$

Therefore:

- If $L_f = \sigma_g = 0$ then the RHS is just $-tS(\mathbf{x})$.
- If $L_f = 0$ then the RHS is $-tS(\mathbf{x}) - \frac{\sigma_g}{2}t(1 - t) \|\mathbf{p_x} - \mathbf{x}\|^2 \leq -tS(\mathbf{x})$.
- If $L_f + \sigma_g > 0$ then the minimum of the RHS is

$$-\frac{b^2}{2a} = -\frac{\|\mathbf{p_x} - \mathbf{x}\|^2}{2(L_f + \sigma_g)} \left( \frac{S(\mathbf{x})}{\|\mathbf{p_x} - \mathbf{x}\|^2} + \frac{\sigma_g}{2} \right)^2,$$

  achieved at

$$t^* = \frac{b}{a} = \frac{1}{L_f + \sigma_g} \left( \frac{S(\mathbf{x})}{\|\mathbf{p_x} - \mathbf{x}\|^2} + \frac{\sigma_g}{2} \right).$$

The adaptive stepsize (14) are defined for the case $L_f + \sigma_g > 0$ is thus:

$$\alpha_k = \min \{1, \alpha_k^*\}, \quad \text{where } \alpha_k^* = \frac{1}{L_f + \sigma_g} \left( \frac{S_k}{r_k} + \frac{\sigma_g}{2} \right). \tag{56}$$

Notice that the RHS of (53) is a quadratic function of $\alpha_k$ with critical point $\alpha_k^*$. If $\alpha_k^* \leq 1$, or equivalently $\frac{\sigma_g}{2} + L_f \geq \frac{S_k}{r_k}$, then $\alpha_k = \alpha_k^*$ and thus (53) becomes

$$F_k - F_{k+1} \geq \alpha_k^* S_k - K(\alpha_k^*) r_k = \frac{r_k}{2(L_f + \sigma_g)} \left( \frac{S_k}{r_k} + \frac{\sigma_g}{2} \right)^2. \tag{57}$$

If $\alpha_k^* > 1$, or equivalently $\frac{\sigma_g}{2} + L_f < \frac{S_k}{r_k}$, then $\alpha_k = 1$ and thus (53) becomes

$$F_k - F_{k+1} \geq S_k - K(1) r_k = S_k - \frac{L_f}{2} r_k. \tag{58}$$

For simplicity and clarity, we will consider separately the two cases: $g$ is strongly convex ($\sigma_g > 0$) or simply convex ($\sigma_g = 0$). Recall that $\Omega$ is the (finite) diameter of $\mathrm{dom}\, g$, and thus we have $r_k \leq \Omega^2 \,\forall k$, a fact that we will be using repeatedly in the sequel. Let $S = \min_{0 \leq i \leq k} S_i$.

### B.2.1 Convex $g$

In this section we consider the case where $g$ is convex but not strongly convex, i.e., $\sigma_g = 0$. Inequality (55) becomes

$$S \leq \frac{\Delta_0 + \frac{L_f}{2} \sum_{i=0}^{k} r_i \alpha_i^2}{\sum_{i=0}^{k} \alpha_i}. \tag{59}$$

**Constant stepsize**  Consider $\alpha_k = \alpha > 0 \,\forall k$. From $r_k \leq \Omega^2$ and (59) we obtain

$$S \leq \frac{\Delta_0}{(k+1)\alpha} + \frac{L_f \Omega^2 \alpha}{2}. \tag{60}$$

The right-hand side converges to $\frac{1}{2} L_f \Omega^2 \alpha$ as $k \to \infty$, i.e., the lower-bound $S$ on the conditional gradient norm converges to within $\frac{1}{2} L_f \Omega^2 \alpha$. It is easy to deduce from the last inequality that $S \leq L_f \Omega^2 \alpha$ within $k \leq \frac{2\Delta_0}{L_f \Omega^2 \alpha^2}$ steps. We conclude that the algorithm converges to an approximate stationary point for the constant stepsize.

**Constant step length**  For a constant step length: $\|\mathbf{x}^{k+1} - \mathbf{x}^k\| = \alpha$. Recall that $\mathbf{x}^{k+1} = \mathbf{x}^k + \alpha_k(\mathbf{p}^k - \mathbf{x}^k)$, the corresponding stepsize is thus $\alpha_k = \frac{\alpha}{\|\mathbf{p}^k - \mathbf{x}^k\|} = \frac{\alpha}{\sqrt{r_k}}$. Inequality (59) becomes

$$S \leq \frac{\Delta_0 + \frac{L_f}{2}(k+1)\alpha^2}{\alpha \sum_{i=0}^{k} \frac{1}{\sqrt{r_k}}} \leq \frac{\Delta_0 + \frac{L_f}{2}(k+1)\alpha^2}{\alpha \frac{k+1}{\Omega}} = \frac{\Delta_0 \Omega}{(k+1)\alpha} + \frac{L_f \Omega \alpha}{2}. \tag{61}$$

Therefore, $S$ converges to within $\frac{L_f \Omega \alpha}{2}$, and $S \leq L_f \Omega \alpha$ within $k \leq \frac{2\Delta_0}{L_f \alpha^2}$ steps. We conclude that the algorithm converges to an approximate stationary point for the stepsizes with constant step length.

**Non-summable but square-summable stepsizes**  Assume that the stepsizes $\alpha_k$ satisfy

$$\sum_{k=0}^{+\infty} \alpha_k = \infty, \qquad \sum_{k=0}^{+\infty} \alpha_k^2 < \infty. \tag{62}$$

A typical example is $\alpha_k = \frac{\alpha}{k+\beta}$, where $\alpha > 0$ and $\beta \geq 0$. This includes the common Frank-Wolfe stepsize $\alpha_k = \frac{2}{k+2}$. From $r_k \leq \Omega^2$ and (59) we obtain

$$S \leq \frac{\Delta_0 + \frac{L_f \Omega^2}{2} \sum_{i=0}^{k} \alpha_i^2}{\sum_{i=0}^{k} \alpha_i}, \tag{63}$$

which clearly converges to 0 as $k \to \infty$. Therefore, the algorithm is guaranteed to converge to a stationary point in this case.

**Diminishing (and non-summable) stepsizes**   Assume that the stepsizes $\alpha_k$ satisfy

$$\sum_{k=0}^{+\infty} \alpha_k = \infty, \qquad \lim_{k \to \infty} \alpha_k = 0. \tag{64}$$

A typical example is $\alpha_k = \frac{\alpha}{\sqrt{k}}$, where $\alpha > 0$. Notice that for any $\epsilon > 0$, we have $\alpha_i^2 < \epsilon \alpha_i$ with $i$ large enough, it is straightforward to show that $\frac{\sum_{i=0}^{k} \alpha_i^2}{\sum_{i=0}^{k} \alpha_i} \to 0$ as $k \to \infty$, and thus (63) implies that $S \to 0$ as well. We conclude that the algorithm is guaranteed to converge to a stationary point.

**Adaptive stepsizes**   This result was obtained previously by Beck [7]. These stepsizes are computed according to (56), which can be simplified as the following for $\sigma_g = 0$:

$$\alpha_k = \min\{1, \alpha_k^*\}, \quad \text{where } \alpha_k^* = \frac{S_k}{L_f r_k}. \tag{65}$$

Then, if $\alpha_k^* \leq 1$, (57) yields

$$F_k - F_{k+1} \geq \frac{S_k^2}{2L_f r_k} \geq \frac{S_k^2}{2L_f \Omega^2}. \tag{66}$$

If $\alpha_k^* > 1$ then (58) and $L_f r_k < S_k$ yield

$$F_k - F_{k+1} \geq S_k - \frac{L_f}{2} r_k \geq \frac{S_k}{2}. \tag{67}$$

Combining the two cases, we obtain

$$F_k - F_{k+1} \geq \frac{S_k}{2} \min\left\{1, \frac{S_k}{L_f \Omega^2}\right\} \geq \frac{S}{2} \min\left\{1, \frac{S}{L_f \Omega^2}\right\}. \tag{68}$$

Therefore,

$$\Delta_0 = F_0 - F^* \geq F_0 - F_{k+1} = \sum_{i=0}^{k}(F_i - F_{i+1}) \geq (k+1)\frac{S}{2} \min\left\{1, \frac{S}{L_f \Omega^2}\right\}, \tag{69}$$

which yields

$$S \leq \max\left\{\frac{2\Delta_0}{k+1}, \frac{\sqrt{2L_f \Omega^2 \Delta_0}}{\sqrt{k+1}}\right\}. \tag{70}$$

Therefore, the algorithm is guaranteed to converge to a stationary point, and the rate of convergence convergence is at least $\mathcal{O}(1/\sqrt{k})$.

### B.2.2   Strongly-convex $g$

In this section we consider the case where $g$ is strongly convex with parameter $\sigma_g > 0$.

Recall from (53) and Lemma 3 that

$$F_k - F_{k+1} \geq \alpha_k S_k - K(\alpha_k) r_k \; \forall k \geq 0, \quad \text{where } K(\alpha) = \frac{1}{2}\alpha\left[(L_f + \sigma_g)\alpha - \sigma_g\right]. \tag{71}$$

Thus if $\alpha_k \leq \frac{\sigma_g}{L_f + \sigma_g}$ we have $K(\alpha_k) \leq 0$ and (71) yields

$$F_k - F_{k+1} \geq \alpha_k S_k. \tag{72}$$

Consider now the case $\alpha_k > \frac{\sigma_g}{L_f + \sigma_g}$ for which $K(\alpha_k) > 0$. From (43) we have $r_k \leq \frac{2S_k}{\sigma_g}$, and thus (71) yields

$$F_k - F_{k+1} \geq \alpha_k S_k - K(\alpha_k)\frac{2S_k}{\sigma_g} = \left(\alpha_k - \frac{2K(\alpha_k)}{\sigma_g}\right) S_k = \alpha_k\left(2 - \frac{L_f + \sigma_g}{\sigma_g}\alpha_k\right) S_k. \tag{73}$$

Combining the two cases, we obtain

$$F_k - F_{k+1} \geq \alpha_k \min\left(1, 2 - \frac{L_f + \sigma_g}{\sigma_g}\alpha_k\right) S_k. \tag{74}$$

If $\alpha_k \geq \frac{2\sigma_g}{L_f + \sigma_g}$ then the RHS of (74) is non-positive, thus this inequality is not helpful. In this case, we can obtain another inequality from (71), noticing that $r_k \leq \Omega^2 \; \forall k$ and $K(\alpha_k) > 0$:

$$F_k - F_{k+1} \geq \alpha_k S_k - K(\alpha_k)\Omega^2 \tag{75}$$

**Constant stepsize**    Assume that $\alpha_k = \alpha \ \forall k \geq 0$. If $0 < \alpha < \frac{2\sigma_g}{L_f + \sigma_g}$ then (74) yields

$$F_k - F_{k+1} \geq \alpha \min\left(1, 2 - \frac{L_f + \sigma_g}{\sigma_g}\alpha\right) S \ \forall k \geq 0. \tag{76}$$

Hence

$$\Delta_0 \geq F_0 - F_{k+1} = \sum_{i=0}^{k}(F_i - F_{i+1}) \geq (k+1)\alpha \min\left(1, 2 - \frac{L_f + \sigma_g}{\sigma_g}\alpha\right) S. \tag{77}$$

We obtain

$$S \leq \frac{\Delta_0}{\alpha \min\left(1, 2 - \frac{L_f + \sigma_g}{\sigma_g}\alpha\right)(k+1)} \quad \forall \alpha < \frac{2\sigma_g}{L_f + \sigma_g}. \tag{78}$$

We conclude that the algorithm is guaranteed to converge to a stationary point with rate of convergence of (at least) $\mathcal{O}(1/k)$ for any $0 < \alpha < \frac{2\sigma_g}{L_f + \sigma_g}$.

For the remaining case $\alpha \geq \frac{2\sigma_g}{L_f + \sigma_g}$, we will derive an upper bound for $S$. Applying (75) we obtain

$$\Delta_0 \geq \sum_{i=0}^{k}(F_i - F_{i+1}) \geq \alpha \sum_{i=0}^{k} S_i - (k+1)K(\alpha)\Omega^2 \geq (k+1)\alpha S - (k+1)K(\alpha)\Omega^2, \tag{79}$$

which yields

$$S \leq \frac{\Delta_0}{\alpha(k+1)} + \frac{K(\alpha)\Omega^2}{\alpha} = \frac{\Delta_0}{\alpha(k+1)} + \frac{1}{2}\left[(L_f + \sigma_g)\alpha - \sigma_g\right]\Omega^2. \tag{80}$$

Therefore, for $\alpha \geq \frac{2\sigma_g}{L_f + \sigma_g}$ the algorithm converges to an approximate stationary point at which the conditional gradient norm is bounded above by $\frac{1}{2}\left[(L_f + \sigma_g)\alpha - \sigma_g\right]\Omega^2$.

**Constant step length**    Consider the stepsize $\alpha_k = \frac{\alpha}{\|\mathbf{p}^k - \mathbf{x}^k\|} = \frac{\alpha}{\sqrt{r_k}}$ for which $\|\mathbf{x}^{k+1} - \mathbf{x}^k\| = \alpha$. Inequality (71) becomes

$$F_k - F_{k+1} \geq \frac{\alpha}{\sqrt{r_k}}S_k - K\left(\frac{\alpha}{\sqrt{r_k}}\right)r_k \tag{81}$$

$$= \frac{\alpha}{\sqrt{r_k}}S_k - \frac{1}{2}\left[(L_f + \sigma_g)\frac{\alpha^2}{r_k} - \sigma_g\frac{\alpha}{\sqrt{r_k}}\right]r_k \tag{82}$$

$$= \frac{\alpha}{\sqrt{r_k}}S_k + \frac{\sigma_g\alpha\sqrt{r_k}}{2} - \frac{1}{2}(L_f + \sigma_g)\alpha^2 \tag{83}$$

$$\geq 2\sqrt{\frac{\alpha}{\sqrt{r_k}}S_k\frac{\sigma_g\alpha\sqrt{r_k}}{2}} - \frac{1}{2}(L_f + \sigma_g)\alpha^2 \tag{84}$$

$$= \alpha\sqrt{2\sigma_g S_k} - \frac{1}{2}(L_f + \sigma_g)\alpha^2. \tag{85}$$

It follows that

$$\Delta_0 \geq (k+1)\alpha\sqrt{2\sigma_g S} - \frac{k+1}{2}(L_f + \sigma_g)\alpha^2 \tag{86}$$

$$\implies \sqrt{S} \leq \frac{\Delta_0}{\alpha\sqrt{2\sigma_g}(k+1)} + \frac{(L_f + \sigma_g)\alpha}{2\sqrt{2\sigma_g}}. \tag{87}$$

For this stepsize scheme, the algorithm converges to an approximate stationary point, within $\frac{(L_f + \sigma_g)^2\alpha^2}{8\sigma_g}$. One can observe that, even though the strong convexity of $g$ still cannot guarantee convergence to a stationary point, it helps improve the bound as well as the rate of convergence from (61).

**Diminishing (and non-summable) stepsizes**   Assume that the stepsizes $\alpha_k$ satisfy

$$\sum_{k=0}^{+\infty} \alpha_k = \infty, \qquad \lim_{k\to\infty} \alpha_k = 0. \tag{88}$$

This scheme also includes the non-summable but square-summable one. Since $\lim_{k\to\infty} \alpha_k = 0$, there exists an integer $k(\omega)$ such that $\alpha_k \leq \omega = \frac{\sigma_g}{L_f + \sigma_g}$ $\forall k \geq k(\omega)$. Now applying (72)

$$\Delta_{k(\omega)} \geq F_{k(\omega)} - F_{k+1} = \sum_{i=k(\omega)}^{k} (F_i - F_{i+1}) \geq \sum_{i=k(\omega)}^{k} \alpha_i S_i \geq \left(\sum_{i=k(\omega)}^{k} \alpha_i\right) S, \tag{89}$$

which yields

$$S \leq \frac{\Delta_{k(\omega)}}{\sum_{i=k(\omega)}^{k} \alpha_i}. \tag{90}$$

Since $(\alpha_k)$ is non-summable, the algorithm converges to a stationary point. Compared to the non-strongly-convex case, we observe that the assumption that $\operatorname{dom} g$ is compact can be relaxed (its diameter $\Omega$ is not used in the proof).

**Adaptive stepsizes**   Recall that the stepsizes in this scheme are given by (56) as

$$\alpha_k = \min\{1, \alpha_k^*\}, \quad \text{where } \alpha_k^* = \frac{1}{L_f + \sigma_g}\left(\frac{S_k}{r_k} + \frac{\sigma_g}{2}\right). \tag{91}$$

If $\alpha_k^* \leq 1$, from (57) and the inequality $(a+b)^2 \geq 4ab$, we obtain

$$F_k - F_{k+1} \geq \frac{r_k}{2(L_f + \sigma_g)} 4 \frac{S_k}{r_k} \frac{\sigma_g}{2} = \frac{\sigma_g S_k}{L_f + \sigma_g}. \tag{92}$$

If $\alpha_k^* > 1$, which is $r_k < \frac{S_k}{\frac{\sigma_g}{2} + L_f}$, then (58) yields

$$F_k - F_{k+1} \geq S_k - \frac{L_f}{2} \frac{S_k}{\frac{\sigma_g}{2} + L_f} = \frac{\sigma_g + L_f}{\sigma_g + 2L_f} S_k \geq \frac{\sigma_g}{\sigma_g + L_f} S_k. \tag{93}$$

Therefore, we always have $F_k - F_{k+1} \geq \omega S_k$ where $\omega = \frac{\sigma_g}{\sigma_g + L_f}$. It then follows that

$$\Delta_0 \geq \sum_{i=0}^{k} (F_i - F_{i+1}) \geq \sum_{i=0}^{k} \omega S_k \geq (k+1)\omega S \implies S \leq \frac{\Delta_0}{\omega(k+1)}. \tag{94}$$

Finally, the line search scheme is guaranteed to achieve the best decrease in the objective, thus the inequality $F_k - F_{k+1} \geq \omega S_k$ also holds and we obtain the same results for this scheme.

### B.2.3   Concave $f$

The results for this case can be obtained in a straightforward manner by setting $L_f = 0$ in the "convex $g$" case.

### B.2.4   Summary of convergence results

We summarize the results in Table 5.

### B.2.5   Convergence of $S(\mathbf{x}^k)$

To complete the proof of Theorem 1, we need to show that for the non-highlighted cases of its table (page 6), we have $\lim_{k\to\infty} S(\mathbf{x}^k) = 0$ and any limit point of the sequence $(\mathbf{x}^k)_{k\geq 0}$ is a stationary point of (9). Indeed, for these cases, $(F_k)_{k\geq 0}$ is a decreasing sequence because $F_k - F_{k+1}$ is bounded below by a non-negative quantity $\delta_k$ (according to Table 5 presented in the previous section). Therefore, $F_k$ is convergent as it is bounded below by $F^*$. Consequently $F_k - F_{k+1} \to 0$, which implies $\delta_k \to 0$ and thus $S_k \to 0$ as well for the considered cases (see Table 5). The results follow in a straightforward manner.

| | stepsize | decrease lower bound $\delta_k$ $(F_k - F_{k+1} \geq \delta_k)$ | optimality upper bound $B_k$ $(\min_{0 \leq i \leq k} S_i \leq B_k)$ |
|---|---|---|---|
| convex $g$ | $\alpha_k = \alpha > 0$ | $\alpha S_k - \frac{L_f \Omega^2 \alpha^2}{2}$ | $\frac{\Delta_0}{(k+1)\alpha} + \frac{L_f \Omega^2 \alpha}{2}$ |
| | $\alpha_k = \frac{\alpha}{\|\mathbf{p}^k - \mathbf{x}^k\|}$ | $\frac{\alpha}{\Omega} S_k - \frac{L_f \alpha^2}{2}$ | $\frac{\Delta_0 \Omega}{(k+1)\alpha} + \frac{L_f \Omega \alpha}{2}$ |
| | $\sum_{k=0}^{+\infty} \alpha_k = \infty$ $\lim_{k\to\infty} \alpha_k = 0$ | $\alpha_k S_k - \frac{L_f \Omega^2 \alpha_k^2}{2}$ | $\frac{\Delta_0}{\sum_{i=0}^k \alpha_i} + \frac{L_f \Omega^2}{2} \frac{\sum_{i=0}^k \alpha_i^2}{\sum_{i=0}^k \alpha_i}$ |
| | adaptive or line search (14) | $\frac{1}{2} \min\left(S_k, \frac{S_k^2}{L_f \Omega^2}\right)$ | $\max\left(\frac{2\Delta_0}{k+1}, \frac{\sqrt{2 L_f \Omega^2 \Delta_0}}{\sqrt{k+1}}\right)$ |
| strongly-convex $g$ | $\alpha_k = \alpha < 2\omega$ | $\alpha \min\left(1, 2 - \frac{\alpha}{\omega}\right) S_k$ | $\frac{\Delta_0}{\alpha \min\left(1, 2 - \frac{\alpha}{\omega}\right)(k+1)}$ |
| | $\alpha_k = \alpha \geq 2\omega$ | $\alpha S_k - K(\alpha)\Omega^2$ | $\frac{\Delta_0}{\alpha(k+1)} + \frac{K(\alpha)}{\alpha}\Omega^2$ |
| | $\alpha_k = \frac{\alpha}{\|\mathbf{p}^k - \mathbf{x}^k\|}$ | $\alpha\sqrt{2\sigma_g S_k} - \frac{1}{2}(L_f + \sigma_g)\alpha^2$ | $\left(\frac{\Delta_0}{\alpha\sqrt{2\sigma_g}(k+1)} + \frac{(L_f+\sigma_g)\alpha}{2\sqrt{2\sigma_g}}\right)^2$ |
| | $\sum_{k=0}^{+\infty} \alpha_k = \infty$ $\lim_{k\to\infty} \alpha_k = 0$ | $\alpha_k \min\left(1, 2 - \frac{\alpha_k}{\omega}\right) S_k$ | $\frac{\Delta_{k(\omega)}}{\sum_{i=k(\omega)}^k \alpha_i}$ |
| | adaptive or line search (14) | $\omega S_k$ | $\frac{\Delta_0}{\omega(k+1)}$ |
| concave $f$ | $\alpha_k = \alpha > 0$ | $\alpha S_k$ | $\frac{\Delta_0}{(k+1)\alpha}$ |
| | $\alpha_k = \frac{\alpha}{\|\mathbf{p}^k - \mathbf{x}^k\|}$ | $\frac{\alpha}{\Omega} S_k$ | $\frac{\Delta_0 \Omega}{(k+1)\alpha}$ |
| | $\sum_{k=0}^{+\infty} \alpha_k = \infty$ | $\alpha_k S_k$ | $\frac{\Delta_0}{\sum_{i=0}^k \alpha_i}$ |
| | adaptive or line search (14) | $\frac{1}{2} S_k$ | $\frac{2\Delta_0}{k+1}$ |

**Table 5:** Summary of convergence analysis of the generalized Frank-Wolfe algorithm. Recall that $\omega = \frac{\sigma_g}{L_f + \sigma_g}$. Whenever a result does not involve $\Omega$, the assumption that $\mathrm{dom}\, g$ being compact can be relaxed.

## C  Proofs of other theoretical results

### C.1  Vanilla Frank-Wolfe fails to learn: the zero-gradient issue

We claimed in §3.1 that vanilla Frank-Wolfe (7) is problematic for learning with SGD because its iterates are piecewise-constant and thus their gradients are zero almost everywhere (more precisely the gradient is undefined on the boundaries while being zero everywhere else). In this section, we present a theoretical justification for this claim.

It suffices to show that $\mathbf{p}^* = \mathrm{argmin}_{\mathbf{p} \in \mathcal{X}} \langle \mathbf{c}, \mathbf{p} \rangle$ is piecewise-constant with respect to $\mathbf{c}$. Let $\Delta_d$ denote the simplex $\{\mathbf{z} \in \mathbb{R}^d \mid \mathbf{1}^\top \mathbf{z} = 1, \mathbf{z} \geq \mathbf{0}\}$. Clearly, the set $\mathcal{X}$ (defined by (5)) can be written as $\{\mathbf{x} \in \mathbb{R}^{nd} \mid \mathbf{x}_i \in \Delta_d \,\forall i \in \mathcal{V}\}$, and thus the above minimization problem can be reduced to solving the following problem for each $i \in \mathcal{V}$ independently:

$$\mathbf{p}_i^* \in \mathrm{argmin}_{\mathbf{p}_i \in \Delta_d} \langle \mathbf{c}_i, \mathbf{p}_i \rangle. \tag{95}$$

For notational convenience, consider the following problem with a constant vector $\mathbf{b} = (b_1, b_2, \ldots, b_d) \in \mathbb{R}^d$:

$$\mathbf{z}^* \in \mathrm{argmin}_{\mathbf{z} \in \Delta_d} \langle \mathbf{b}, \mathbf{z} \rangle. \tag{96}$$

Let $s^*$ be the index of the minimum element of $\mathbf{b}$, i.e., $s^* = \mathrm{argmin}_s b_s$. Let $\mathbf{e}_s \in \{0, 1\}^d$ denote the one-hot vector where the $s^{\text{th}}$ element is one. We have:

$$\langle \mathbf{b}, \mathbf{z} \rangle = \sum_{s=1}^d b_s z_s \geq \sum_{s=1}^d b_{s^*} z_s = b_{s^*} \sum_{s=1}^d z_s = b_{s^*} = \langle \mathbf{b}, \mathbf{e}_{s^*} \rangle \quad \forall \mathbf{z} \in \Delta_d. \tag{97}$$

Therefore, $\mathbf{e}_{s^*}$ is an optimal solution to (96). It is straightforward that the index of the minimum element of a vector is piecewise constant, thus $\mathbf{e}_{s^*}$ is also piecewise constant (as a function of $\mathbf{b}$). Therefore, $\mathbf{e}_{s^*}$ is not continuous (thus non-differentiable) on the boundaries, while in the constant regions, its gradient is zero.

**Remark.** We can deduce that the iteration complexity of vanilla Frank-Wolfe is $\mathcal{O}(nd)$.

## C.2 Proof of the relaxation tightness (Theorem 2)

We give a proof of Theorem 2 in §4.2. Recall that we have to prove $E^* \leq E(\bar{\mathbf{x}}_r^*) \leq E^* + M - m + C$, where

$$C = \begin{cases} \sqrt{n\left(1 - \frac{1}{d}\right)} \left(\|\mathbf{u}\|_2 + \sqrt{n}\,\|\mathbf{P}\|_2\right) & \text{for nearest rounding} \\ 0 & \text{for BCD rounding.} \end{cases} \tag{98}$$

Let $\mathbf{x}^*$ be such that $E(\mathbf{x}^*) = E^*$ and consider first the BCD rounding scheme. As this scheme is guaranteed to not increase the energy, we have

$$E(\bar{\mathbf{x}}_r^*) \leq E(\mathbf{x}_r^*) = E_r(\mathbf{x}_r^*) - r(\mathbf{x}_r^*) \leq E_r(\mathbf{x}^*) - r(\mathbf{x}_r^*) = E(\mathbf{x}^*) + r(\mathbf{x}^*) - r(\mathbf{x}_r^*) \leq E^* + M - m.$$

It remains to prove the result for the nearest rounding scheme. In this scheme, the discrete energy may increase (or decrease), but it can be shown that the variation is bounded by the given constant: $|E(\bar{\mathbf{x}}_r^*) - E(\mathbf{x}_r^*)| \leq C$. Then, the rest of the proof is similar to the BCD case. This bounding inequality is proved as follows.

Suppose that we obtain a discrete solution $\mathbf{y} \in \mathcal{X} \cap \{0,1\}^{nd}$ from some $\mathbf{x} \in \mathcal{X}$ using nearest rounding. We will prove that

$$|E(\mathbf{x}) - E(\mathbf{y})| \leq C, \tag{99}$$

where

$$C = \sqrt{n\left(1 - \frac{1}{d}\right)} \left(\|\mathbf{u}\|_2 + \sqrt{n}\,\|\mathbf{P}\|_2\right). \tag{100}$$

**Lemma 4.** *For any $\mathbf{z} \in \Delta_d$ (see §C.1 for notation) and its rounded vector $\mathbf{v} \in \Delta_d \cap \{0,1\}^d$, i.e., $v_i = 1$ if $i = \mathrm{argmax}_{1 \leq j \leq d}\, v_j$ and $v_j = 0\ \forall j \neq i$, we have*

$$\|\mathbf{z} - \mathbf{v}\|_2^2 \leq 1 - \frac{1}{d}. \tag{101}$$

*Proof.* Without loss of generality, assume that $z_1$ is the maximum element of $\mathbf{z}$. Then, we have $v_1 = 1$ and $v_j = 0\ \forall j > 1$.

$$\|\mathbf{z} - \mathbf{v}\|_2^2 = \sum_{i=1}^d (z_i - v_i)^2 = (z_1 - 1)^2 + \sum_{i=1}^d z_i^2 = S^2 + z_2^2 + \cdots + z_d^2, \tag{102}$$

where $S = z_2 + \cdots + z_d$. We will make use of the following trivial inequality:

$$z_i + S \leq 1 \quad \forall i \geq 2. \tag{103}$$

On one hand, summing the $d - 1$ inequalities (103) (for $i = 2, \ldots, d$) we obtain

$$S + (d-1)S \leq d - 1 \implies S \leq 1 - \frac{1}{d}. \tag{104}$$

On the other hand, multiplying (103) with $z_i$ and summing up the obtained $d - 1$ inequalities we get

$$\sum_{i=2}^d z_i^2 + S^2 \leq S \tag{105}$$

Finally, from (102), (104), and (105) we get (101). $\qquad\square$

Back to (99). Applying (101) we have

$$\|\mathbf{x} - \mathbf{y}\|_2^2 = \sum_{i=1}^n \|\mathbf{x}_i - \mathbf{y}_i\|_2^2 \leq n\left(1 - \frac{1}{d}\right). \tag{106}$$

On the other hand

$$\|\mathbf{x} + \mathbf{y}\|_2^2 = \sum_{i=1}^n \|\mathbf{x}_i + \mathbf{y}_i\|_2^2 \leq \sum_{i=1}^n \left[\mathbf{1}^\top (\mathbf{x}_i + \mathbf{y}_i)\right]^2 = 4n. \tag{107}$$

Applying the two above inequalities, together with the triangle and Cauchy-Schwarz inequalities we have:

$$
\begin{aligned}
|E(\mathbf{x}) - E(\mathbf{y})| &= \left| \mathbf{u}^\top (\mathbf{x} - \mathbf{y}) + \frac{1}{2}\mathbf{x}^\top \mathbf{P}\mathbf{x} - \frac{1}{2}\mathbf{y}^\top \mathbf{P}\mathbf{y} \right| \\
&\leq \left| \mathbf{u}^\top (\mathbf{x} - \mathbf{y}) \right| + \frac{1}{2} \left| \mathbf{x}^\top \mathbf{P}\mathbf{x} - \mathbf{y}^\top \mathbf{P}\mathbf{y} \right| \\
&= \left| \mathbf{u}^\top (\mathbf{x} - \mathbf{y}) \right| + \frac{1}{2} \left| (\mathbf{x} - \mathbf{y})^\top \mathbf{P}(\mathbf{x} + \mathbf{y}) \right| \\
&\leq \|\mathbf{u}\|_2 \|\mathbf{x} - \mathbf{y}\|_2 + \frac{1}{2} \|\mathbf{x} - \mathbf{y}\|_2 \|\mathbf{P}\|_2 \|\mathbf{x} + \mathbf{y}\|_2 \\
&\leq \|\mathbf{u}\|_2 \sqrt{n\left(1 - \frac{1}{d}\right)} + \frac{1}{2}\sqrt{n\left(1 - \frac{1}{d}\right)} \|\mathbf{P}\|_2 \, 2\sqrt{n} \\
&= C.
\end{aligned}
$$

This completes the proof.

## D   Implementation details of all methods

We present the implementation details for all the methods presented in the experiments (§5). Recall that our problem of interest is

$$
\min_{\mathbf{x} \in \mathcal{X}} E(\mathbf{x}) \triangleq \frac{1}{2}\mathbf{x}^\top \mathbf{P}\mathbf{x} + \mathbf{u}^\top \mathbf{x},
$$

and that the same initialization $\mathbf{x}^0 = \mathrm{softmax}(-\mathbf{u})$ is used for all methods.

**High-dimensional filtering for gradient computation**   For all methods, we need to compute the energy gradient $\nabla E(\mathbf{x}) = \mathbf{P}\mathbf{x} + \mathbf{u}$ at each iteration, where the evaluation of $\mathbf{P}\mathbf{x}$ is an expensive operation because the graph is fully-connected (i.e., $\mathbf{P}$ is dense). Fortunately, since the pairwise potentials are Gaussian, this multiplication can be performed efficiently (and approximately) in $\mathcal{O}(nd)$ time using high-dimensional filtering, which is the key idea behind the original dense CRFs paper [34]. We refer to this reference for more details. Our code is based on the efficient GPU implementation of Monteiro et al. [54].[10]

### D.1   Euclidean Frank-Wolfe ($\ell_2$FW)

We give the details for the main update (11) of Euclidean Frank-Wolfe as presented in §3.3. This step follows from

$$
\forall k \geq 0 : \quad \mathbf{p}^k = \operatorname*{argmin}_{\mathbf{p} \in \mathcal{X}} \left\{ \langle \mathbf{P}\mathbf{x}^k + \mathbf{u}, \mathbf{p} \rangle + \frac{\lambda}{2} \|\mathbf{p}\|_2^2 \right\} \tag{108}
$$

$$
= \operatorname*{argmin}_{\mathbf{p} \in \mathcal{X}} \left\{ \frac{\lambda}{2} \left\| \mathbf{p} + \frac{1}{\lambda}(\mathbf{P}\mathbf{x}^k + \mathbf{u}) \right\|^2 \right\} \tag{109}
$$

$$
= \Pi_{\mathcal{X}} \left( -\frac{1}{\lambda}(\mathbf{P}\mathbf{x}^k + \mathbf{u}) \right). \tag{110}
$$

Recall that $\Pi_{\mathcal{X}}(\mathbf{v})$ denotes the projection of a vector $\mathbf{v}$ onto the set $\mathcal{X}$. Recall also from (5) that $\mathcal{X} = \left\{ \mathbf{x} \in \mathbb{R}^{nd} : \mathbf{x} \geq \mathbf{0}, \mathbf{1}^\top \mathbf{x}_i = 1 \; \forall i \in \mathcal{V} \right\}$, thus the projection on $\mathcal{X}$ clearly reduces to $n$ *independent* projections onto the probability simplex $\Delta_d = \left\{ \mathbf{z} \in \mathbb{R}^d : \mathbf{1}^\top \mathbf{z} = 1, \mathbf{z} \geq \mathbf{0} \right\}$ for each $i \in \mathcal{V}$. Projection onto the simplex is a rather well studied problem in the literature [18], and we present below the solution (that also shows how we implemented this operation).

**Lemma 5.** *For a given vector $\mathbf{c} \in \mathbb{R}^d$, the optimal solution $\mathbf{z}^*$ to*

$$
\min_{\mathbf{1}^\top \mathbf{z} = 1, \mathbf{z} \geq \mathbf{0}} \|\mathbf{z} - \mathbf{c}\|^2 \tag{111}
$$

---

[10] https://github.com/MiguelMonteiro/permutohedral_lattice

*is given as follows. Sort $\mathbf{c}$ in decreasing order to obtain a vector $\mathbf{a} = (a_1, a_2, \ldots, a_d)$ (i.e., $a_1 \geq a_2 \geq \cdots \geq a_d$) and let*

$$\gamma_k = \frac{1}{k}(a_1 + a_2 + \cdots + a_k - 1), \quad k = 1, 2, \ldots, d. \tag{112}$$

*Let $k^*$ be the largest $k$ such that $a_k > \gamma_k$, then the optimal solution is given by*

$$\mathbf{z}^* = \max(\mathbf{c} - \gamma_{k^*}, 0). \tag{113}$$

In the above, the "max" and "$-$" operations are understood to be element-wise.

**Remark.** If we use an $\mathcal{O}(d \log d)$ sorting algorithm, then we see that the per-iteration complexity of Euclidean Frank-Wolfe is $\mathcal{O}(nd \log d)$. It should be noted, however, that highly-efficient simplex-projection algorithms exist and have $\mathcal{O}(d)$ complexity in practice [18, Table 1], yielding $\mathcal{O}(nd)$ complexity, which is the same as in vanilla Frank-Wolfe (see §C.1).

### D.2 Entropic Frank-Wolfe ($e$FW)

We give the details for the main update (12) of Entropic Frank-Wolfe as presented in §3.3. We need to show that

$$\mathbf{p}^k = \underset{\mathbf{p} \in \mathcal{X}}{\arg\min} \left\{ \langle \mathbf{P}\mathbf{x}^k + \mathbf{u}, \mathbf{p} \rangle - \lambda H(\mathbf{p}) \right\} = \text{softmax}\left( -\frac{1}{\lambda}(\mathbf{P}\mathbf{x}^k + \mathbf{u}) \right) \quad \forall k \geq 0, \tag{114}$$

where $H(\mathbf{x}) = -\sum_{i \in \mathcal{V}} \sum_{s \in \mathcal{S}} x_{is} \log x_{is}$. Again, the above reduces to $n$ independent subproblems over each $i \in \mathcal{V}$ to which the solutions are given by the following lemma.

**Lemma 6.** *For a given vector $\mathbf{c} \in \mathbb{R}^d$, the optimal solution $\mathbf{z}^*$ to*

$$\min_{\mathbf{1}^\top \mathbf{z} = 1, \mathbf{z} \geq 0} \left\{ \langle \mathbf{c}, \mathbf{z} \rangle + \sum_{s=1}^{d} z_s \log z_s \right\} \tag{115}$$

*is $\mathbf{z}^* = \text{softmax}(-\mathbf{c})$.*

*Proof.* The Lagrangian of the above problem is given by

$$L(\mathbf{z}, \boldsymbol{\mu}, \nu) = \langle \mathbf{c}, \mathbf{z} \rangle + \sum_{s=1}^{d} z_s \log z_s + \boldsymbol{\mu}^\top(-\mathbf{z}) + \nu(\mathbf{1}^\top \mathbf{z} - 1) \tag{116}$$

$$= -\nu + \sum_{s=1}^{d} (c_s z_s + z_s \log z_s - \mu_s z_s + \nu z_s), \tag{117}$$

where $\boldsymbol{\mu} = (\mu_1, \mu_2, \ldots, \mu_d) \geq \mathbf{0}$ and $\nu \in \mathbb{R}$ are the Lagrange multipliers.

Observe that the given problem is convex and the corresponding Slater's constraint qualification holds (i.e., there exists $\mathbf{z} \in \mathbb{R}^d$ such that $\mathbf{1}^\top \mathbf{z} = 1$ and $\mathbf{z} > \mathbf{0}$), it suffices to solve the following Karush–Kuhn–Tucker (KKT) system to obtain the optimal solution:

$$\frac{\partial L(\mathbf{z}, \boldsymbol{\mu}, \nu)}{\partial z_s} = c_s + \log z_s + 1 - \mu_s + \nu = 0 \quad \forall 1 \leq s \leq d, \tag{118}$$

$$\mathbf{1}^\top \mathbf{z} = 1, \tag{119}$$

$$\mathbf{z} \geq \mathbf{0}, \tag{120}$$

$$\boldsymbol{\mu} \geq \mathbf{0}, \tag{121}$$

$$\mu_s z_s = 0 \quad \forall 1 \leq s \leq d. \tag{122}$$

The first equation implies $z_s > 0 \; \forall s$, and thus in combination with the last, we obtain $\mu_s = 0 \; \forall s$. Therefore, the first equation becomes

$$z_s = \exp(-1 - \nu) \exp(-c_s) \; \forall s. \tag{123}$$

Summing up this result for all $s$, and taking into account the second equation, we obtain

$$\exp(-1 - \nu) = \frac{1}{\sum_{s=1}^{d} \exp(-c_s)}. \tag{124}$$

Combining (123) and (124) we obtain

$$z_s = \frac{\exp(-c_s)}{\sum_{t=1}^{d} \exp(-c_t)} \quad \forall 1 \le s \le d. \tag{125}$$

In other words, $\mathbf{z} = \mathrm{softmax}(-\mathbf{c})$. $\qquad\square$

**Remark.** It is clear that the per-iteration complexity of Entropic Frank-Wolfe is $\mathcal{O}(nd)$, which is the same as in vanilla Frank-Wolfe.

### D.3 Projected gradient descent (PGD)

This algorithm consists in the following updates:

$$\mathbf{p}^k = \Pi_{\mathcal{X}}(\mathbf{x}^k - \nabla E(\mathbf{x})), \qquad \mathbf{x}^{k+1} = \mathbf{x}^k + \alpha_k(\mathbf{p}^k - \mathbf{x}^k), \tag{126}$$

where the stepsize $\alpha_k$ follows one of the schemes presented in §4.1. It is worth noting that this variant of PGD is eligible to exact line search (14). We observe in our experiments that using the line search scheme produces the same results as setting $\alpha_k = 1$. Thus we used this constant scheme for both training and prediction. The same applies to the Frank-Wolfe variants.

### D.4 Fast proximal gradient method (PGM)

The original PGM [48] consists in updating

$$\mathbf{x}^{k+1} = \underset{\mathbf{x} \in \mathcal{X}}{\mathrm{argmin}} \left\{ \langle \nabla E(\mathbf{x}^k), \mathbf{x} \rangle + \frac{1}{2\alpha_k} \|\mathbf{x} - \mathbf{x}^k\|_2^2 \right\}, \tag{127}$$

which can be re-written as

$$\mathbf{x}^{k+1} = \Pi_{\mathcal{X}}(\mathbf{x}^k - \alpha_k \nabla E(\mathbf{x}^k)). \tag{128}$$

The above is precisely another variant of PGD (which is not eligible to exact line search (14)). While this algorithm is also supported by our implementation, the results presented in §5 are obtained using another variant called the *fast* PGM, also known as FISTA [9]. This algorithm consists in the following updates, where $\mathbf{y}^0 = \mathbf{x}^0 = \mathrm{softmax}(-\mathbf{u})$ and $t_0 = 1$:

$$\mathbf{x}^{k+1} = \Pi_{\mathcal{X}}(\mathbf{y}^k - \alpha_k \nabla E(\mathbf{y}^k)), \tag{129}$$

$$t_{k+1} = \frac{1 + \sqrt{1 + 4t_k^2}}{2}, \tag{130}$$

$$\mathbf{y}^{k+1} = \mathbf{x}^{k+1} + \frac{t_k - 1}{t_{k+1}}(\mathbf{x}^{k+1} - \mathbf{x}^k). \tag{131}$$

While the optimal value of $\alpha_k$ can be determined using *backtracking* [7], this process is very expensive as it requires evaluating the energy many times. Therefore, in practice, we use the constant scheme $\alpha_k = \alpha \in [0, 1]$. Doing a grid search on a random subset of 10 validation images, we found (again) that $\alpha_k = 1$ is the best, and thus it is used for all the experiments.

### D.5 Entropic mirror descent (EMD)

Mirror descent (MD) [8, 55] is a generalization of PGM to a more general distance function. Each iteration of MD takes the following form:

$$\mathbf{x}^{k+1} = \underset{\mathbf{x} \in \mathcal{X}}{\mathrm{argmin}} \left\{ \langle \nabla E(\mathbf{x}^k), \mathbf{x} \rangle + \frac{1}{\alpha_k} B_\phi(\mathbf{x}, \mathbf{x}^k) \right\}, \tag{132}$$

where $\phi : \mathcal{X} \to \mathbb{R}$ is a convex and continuously differentiable function on the interior of $\mathcal{X}$, and $B_\phi : \mathcal{X} \times \mathcal{X} \to \mathbb{R}$ is its associated Bregman divergence, defined by

$$B_\phi(\mathbf{x}, \mathbf{y}) = \phi(\mathbf{x}) - \phi(\mathbf{y}) - \langle \nabla \phi(\mathbf{y}), \mathbf{x} - \mathbf{y} \rangle. \tag{133}$$

Clearly, for $\phi(\mathbf{x}) = \frac{1}{2} \|\mathbf{x}\|_2^2$ we recover the PGM update (127). We provide an implementation for the so-called *entropic* variant of mirror descent [8], corresponding to choosing $\phi$ to be the negative entropy:

$$\phi(\mathbf{x}) = \sum_{i=1}^{n} \sum_{s=1}^{d} x_{is} \log x_{is}. \tag{134}$$

With this choice of $\phi$, it is easy to check that the Bregman divergence (133) becomes the following so-called Kullback-Leibler divergence:

$$B_{\mathrm{KL}}(\mathbf{x}, \mathbf{y}) = \sum_{i=1}^{n} \sum_{s=1}^{d} x_{is} \log \frac{x_{is}}{y_{is}}. \tag{135}$$

The MD update (132) thus becomes

$$\mathbf{x}^{k+1} = \operatorname*{argmin}_{\mathbf{x} \in \mathcal{X}} \left\{ \langle \alpha_k \nabla E(\mathbf{x}^k) - \log \mathbf{x}^k, \mathbf{x} \rangle + \sum_{i=1}^{n} \sum_{s=1}^{d} x_{is} \log x_{is} \right\}, \tag{136}$$

where the $\log$ operation is taken element-wise. According to Lemma 6 (§D.2), we obtain

$$\mathbf{x}^{k+1} = \operatorname{softmax} \left( \log \mathbf{x}^k - \alpha_k \nabla E(\mathbf{x}^k) \right). \tag{137}$$

Let $\mathbf{g}^k$ denote the gradient $\nabla E(\mathbf{x}^k)$, the above reads

$$x_{is}^{k+1} = \frac{x_{is}^k \exp(-\alpha_k g_{is}^k)}{\sum_{t=1}^{d} x_{it}^k \exp(-\alpha_k g_{it}^k)} \quad \forall i \in \mathcal{V}, \forall s \in \mathcal{S}. \tag{138}$$

**Numerically stable EMD** In practice, the above expression of $\mathbf{x}^{k+1}$ may lead to numerical underflow or overflow. We overcome this by using the following modified iterate:

$$x_{is}^{k+1} = \frac{(x_{is}^k + \epsilon) \exp(-\alpha_k g_{is}^k + m_i^k)}{\sum_{t=1}^{d} (x_{it}^k + \epsilon) \exp(-\alpha_k g_{it}^k + m_i^k)} \quad \forall i \in \mathcal{V}, \forall s \in \mathcal{S}, \tag{139}$$

where $\epsilon = 10^{-10}$ and $m_i^k = \alpha_k \min_{1 \le s \le d} g_{is}^k \ \forall i \in \mathcal{V}$.

### D.6 Alternating direction method of multipliers (ADMM)

The nonconvex ADMM for MAP inference [41] consists in the following updates, where $\mathbf{z}^0 = \operatorname{softmax}(-\mathbf{u})$ and $\mathbf{y}^0 = \mathbf{0}$:

$$\mathbf{x}^{k+1} = \Pi_{\mathcal{X}} \left( \mathbf{z}^k - \frac{1}{\rho} (\mathbf{y}^k + \frac{1}{2} \mathbf{P} \mathbf{z}^k + \mathbf{u}) \right), \tag{140}$$

$$\mathbf{z}^{k+1} = \Pi_{\mathcal{X}} \left( \mathbf{x}^{k+1} - \frac{1}{\rho} (-\mathbf{y}^k + \frac{1}{2} \mathbf{P} \mathbf{x}^{k+1}) \right), \tag{141}$$

$$\mathbf{y}^{k+1} = \mathbf{y}^k + \rho (\mathbf{x}^{k+1} - \mathbf{z}^{k+1}). \tag{142}$$

We refer to the original paper [41] for more details. In our experiments, we set $\rho = 1$ for simplicity. Since the expensive computation $\mathbf{Px}$ are done two times in each ADMM iteration (one in (140), another in (141)), this algorithm is roughly two times slower than the others. For a fair comparison, in our implementation we view (140) and (141) as two separate iterations (note that both $\mathbf{x}^{k+1}$ and $\mathbf{z}^{k+1}$ are feasible points).

Finally, we should note that the adaptive scheme for the penalty parameter $\rho$ proposed by Lê-Huu and Paragios [41] is not applicable to our case, as we use only 5 iterations in our experiments (which is equivalent to only 2.5 regular iterations due to our above iteration separation).

## E  Detailed experimental setup and environment

### E.1  CNN-CRF architectures

**CNN-CRF** Our segmentation model is a standard combination of a CNN and a CRF [76]. Given an input image $\mathbf{Z} \in \mathbb{R}^{H \times W \times 3}$, the CNN produces an output $\mathbf{Y} \in \mathbb{R}^{H \times W \times K}$ (where $K$ is the number of object classes) called the *logits*, which is then fed into the CRF to produce a final output $\mathbf{X} \in \mathbb{R}^{H \times W \times K}$:

$$\mathbf{Y} = \operatorname{CNN}(\mathbf{Z}; \boldsymbol{\theta}^u), \quad \mathbf{X} = \operatorname{CRF}(\mathbf{Y}; \boldsymbol{\theta}^p), \tag{143}$$

where $\boldsymbol{\theta}^u$ and $\boldsymbol{\theta}^p$ are (typically trainable) parameters. The prediction is then obtained by taking the $\operatorname{argmax}$ along the last dimension of $\mathbf{X}$. For the CNN part, we consider two strong architectures: DeepLabv3 with ResNet101 backbone [16], and DeepLabv3+ with Xception65 backbone [17]. The reader is referred to the corresponding references for further details.

**Dense CRF** The CRF is defined over the input image such that each pixel is a node, and its labels are the object classes. Thus, using the notation defined in §2.1, we have $n = HW$, $d = K$, and $\mathcal{S} = \{1, 2, \ldots, K\}$. The CRF produces $\mathbf{X}$ in (143) by minimizing the energy (6) with appropriately constructed potentials, and then simply reshaping the solution $\mathbf{x} \in \mathbb{R}^{HWK}$ into $H \times W \times K$. During training we skip the rounding step in CRF inference, so that the returned $\mathbf{x}$ is real-valued, which is more suitable for learning with the standard cross-entropy loss function. The unary potentials $\mathbf{u}$ is defined by to be the additive inverse of the logits $\mathbf{Y}$, reshaped correctly: $\mathbf{u} = -\text{vec}(\mathbf{Y})$, where vec denotes the flattening operator. We use the fully-connected model introduced by Krähenbühl and Koltun [34] in which any pair of pixels $(i, j)$ is an edge with a pairwise potential of the form $\theta_{ij}(s, t) = \mu(s, t)k(\mathbf{f}_i, \mathbf{f}_j)\ \forall s, t \in \mathcal{S}$, where $\mu : \mathcal{S} \times \mathcal{S} \to \mathbb{R}$ is the so-called label compatibility function, and $k$ is a Gaussian kernel over some image features $(\mathbf{f}_i, \mathbf{f}_j)$. For a pixel $i$, we use its position $\mathbf{p}_i \in \mathbb{N}^2$ and its color $\mathbf{c}_i \in [0, 255]^3$ as features, and define the kernel as

$$k(\mathbf{f}_i, \mathbf{f}_j) = w_1 \exp\left(-\frac{\|\mathbf{p}_i - \mathbf{p}_j\|_2^2}{2\alpha^2} - \frac{\|\mathbf{c}_i - \mathbf{c}_j\|_2^2}{2\beta^2}\right) + w_2 \exp\left(-\frac{\|\mathbf{p}_i - \mathbf{p}_j\|_2^2}{2\gamma^2}\right) \quad \forall i, j \in \mathcal{V}, \tag{144}$$

where $w_1, w_2$ are learnable kernel weights, and $\alpha, \beta, \gamma$ are hyperparameters. Following Zheng et al. [76], we use class-dependent kernel weights to increase the number of trainable parameters. Unlike Zheng et al. [76], for simplicity we use the default values $\alpha = 80, \beta = 13, \gamma = 3$ set by Krähenbühl and Koltun [34] in all experiments, instead of doing a cross validation to find the best values. Finally, we use the Potts compatibility function: $\mu(s, t) = w\mathbb{1}_{[s \neq t]}$ with $w = 1$ for the inference experiments in §5.2, and also for CRF initialization in the learning experiments in §5.3.

## E.2 Datasets

We provide further details on the datasets. PASCAL VOC [22] contains 4369 images of 21 classes, split into 1464 (*train*), 1449 (*val*), and 1456 (*test*) image subsets. As a standard practice, we augment the dataset with images from Hariharan et al. [27], resulting in 10 582 training images (*trainaug*). Cityscapes [19] contains 5000 images of 19 classes, split into 2975 (*train*), 500 (*val*), and 1525 (*test*) image subsets. In addition, it also provides 19 998 coarsely annotated images (*train_extra*). We report the performance in terms of mIoU across the semantic classes (21 for PASCAL VOC and 19 for Cityscapes).

## E.3 CNN training recipes

To fully train DeepLabv3 and DeepLabv3+, we follow closely the published recipes [16, 17] for this task. Below we present the most important information, and refer to the references for further details.

We first pretrain DeepLabv3 and DeepLabv3+ on the COCO [46] dataset (by selecting only the images that contain the classes defined in PASCAL VOC), and then finetune them on PASCAL VOC (*trainaug*) and Cityscapes (*train*). During training, we apply data augmentation by (randomly) left-right flipping, scaling the input images (from 0.5 to 2.0), and cropping (with crop size of $513 \times 513$ for PASCAL VOC and $769 \times 769$ for Cityscapes). We employ a *poly* learning rate schedule: $\eta_m = \eta_0 \left(1 - \frac{m}{M}\right)^p$, where $\eta_0$ is the initial learning rate, $m$ is the step counter, and $M$ is the total number of training steps. For all trainings, we set $p = 0.9$ and $\eta_0 = 0.001$ (except $\eta_0 = 0.01$ for pretraining on COCO), and a batch size of 16 images. The value of $M$ is calculated from the number of training epochs, which we set to be 50 for COCO pretraining, 50 for finetuning on PASCAL VOC, and 500 for finetuning on Cityscapes. We should note some differences compared to the original papers [16, 17]. In particular, they did not specify the learning rate and number of steps for COCO pretraining. Furthermore, they used $\eta_0 = 0.0001$, which did not yield better results than $\eta_0 = 0.001$ in our implementation. Finally, in terms of the number of epochs, we used similar values to theirs. Indeed, they set 30 000 and 90 000 training steps for the 10 582 and 2975 training images of PASCAL VOC and Cityscapes, respectively. With a batch size of 16, these are equivalent to 45 and 484 epochs. Table 6 shows that our obtained results are similar to previous work [16, 17].

## E.4 Training time and memory footprint

Our experiments are performed on a Linux server of 4 Nvidia V100 GPUs, using PyTorch 1.7. With a batch size of 16 on 4 GPUs (i.e., 4 images per GPU), both DeepLabv3 and DeepLabv3+ take

| Dataset | Model | Published [16, 17] | Reproduced |
|---------|-------|--------------------|------------|
| VOC | DeepLabv3 | 78.51 | 81.83 |
|     | DeepLabv3+ | 82.45 | 82.89 |
| Cityscapes | DeepLabv3 | 77.82 | 76.73 |
|            | DeepLabv3+ | 79.14 | 79.55 |

**Table 6:** Performance of our reproduced DeepLab models compared to the original papers [16, 17]. The mIoU scores are obtained on the *val* sets, without test time augmentation.

~7min/epoch on PASCAL VOC (with $513 \times 513$ crops). Thanks to our efficient GPU implementation (which will be made publicly available), plugging in the (5-step) CRF only increases that to ~9 minutes ($1.2$–$1.3\times$ slower) for all inference methods (here we should note that CRF's running time is dominated by computing $\mathbf{Px}$ at each step, which is why the running is similar across the methods). In terms of memory usage, DeepLabv3+ takes ~27.7GB (per GPU for $4$ images) while DeepLabv3 takes ~15.6GB. We found that the additional memory usage of the CRF (which has only $1323$ trainable parameters) are negligible for the Frank-Wolfe variants as well as for PGD, while PGM and ADMM require an extra amount of ~300MB (probably due to the additional storage of the variable $\mathbf{y}$ at each iteration, see §D).

# F    Additional results

## F.1    Detailed results on the test sets

The detailed results on the test sets can be found on the corresponding submission websites whose URLs are given in Table 7.

| Model | PASCAL VOC | Cityscapes |
|-------|------------|------------|
| DeepLabv3+ [17] | 87.8 | 82.1 |
| DeepLabv3+ (this work) | 87.6[1] | 83.5[3] |
| DeepLabv3+ with $\ell_2$FW CRF | **88.0**[2] | 83.6[4] |

[1] http://host.robots.ox.ac.uk:8080/anonymous/BUXULK.html

[2] http://host.robots.ox.ac.uk:8080/anonymous/YFJJLW.html

[3] https://www.cityscapes-dataset.com/anonymous-results/?id=845bd062fddae249ec0f4987d30f2f9be6e6716654513e7e6733d3f56e976532

[4] https://www.cityscapes-dataset.com/anonymous-results/?id=84e788da7c55eeeb4840b70407ed665006494c99e5d34e0bd8704d66d9c8b864

**Table 7:** Performance on the *test* sets.

## F.2    Results for trainable $\alpha_k$ and $\lambda$

We carried out an experiment with $\ell_2$FW and $e$FW ($\lambda = 0.7$) in which we allow the stepsize $\alpha_k$ at each CRF iteration to be learnable (initialized at $0.5$). We observe the stepsizes at all the steps behave very similarly (i.e., increasing or decreasing together). In addition, for $e$FW they tend to increase during training, while for $\ell_2$FW they tend to decrease. In addition, we also tried setting the regularization weight $\lambda$ to trainable. We initialized it at $1.0$ for $\ell_2$FW and at $0.7$ for $e$FW. For both solvers, we found that $\lambda$ increased during training. Regarding accuracy, we did not observe significant differences compared to fixed $\alpha_k$ and fixed $\lambda$, although tuning the learning rates specifically for these variables could potentially lead to improved performance. See Table 8 for the details.

## F.3    Results for fined-grained analysis

We randomly picked a trained checkpoint among the different runs for DeepLabv3+ with $\ell_2$FW ($\lambda = 1.0$) and DeepLabv3+ with $e$FW ($\lambda = 0.7$), and evaluated them using 5, 10, and 25 CRF iterations on the PASCAL VOC validation set. The results are shown in Table 9.

## F.4    Additional inference results

| Regularizer | $\lambda$ | | Stepsize | mIoU |
|---|---|---|---|---|
| $\ell_2$ | 1.0 fixed | | 1.0 fixed | 0.8490489721 |
| | 1.0 fixed | | 0.5 fixed | 0.849458456 |
| | 1.0 fixed | | 0.5 learnable | 0.849185586 |
| | 1.0 learnable | | 0.5 learnable | 0.8492224813 |
| Entropy | 0.7 fixed | | 1.0 fixed | 0.8495011926 |
| | 0.7 fixed | | 0.5 fixed | 0.8493972421 |
| | 0.7 fixed | | 0.5 learnable | 0.849845171 |
| | 0.7 learnable | | 0.5 learnable | 0.8491678238 |

**Table 8:** Comparison between trainable and fixed $\lambda$ and $\alpha_k$

| Method | Steps | mIoU | background | aeroplane | bicycle | bird | boat | bottle | bus | car | cat | chair | cow | diningtable | dog | horse | motorbike | person | pottedplant | sheep | sofa | train | tvmonitor |
|---|---|---|---|---|---|---|---|---|---|---|---|---|---|---|---|---|---|---|---|---|---|---|---|
| CNN | | 82.89 | 95.79 | 91.80 | 44.89 | 89.92 | 71.49 | 83.54 | 94.68 | 91.54 | 95.42 | 52.36 | 95.51 | 70.25 | 93.63 | 93.08 | 88.27 | 90.20 | 68.03 | 92.62 | 66.95 | 92.33 | 78.45 |
| $\ell_2$FW | 5 | 85.51 | 96.66 | 93.56 | 60.56 | 90.47 | 80.23 | 83.51 | 96.94 | 91.68 | 95.38 | 54.92 | 95.87 | 76.11 | 94.01 | 93.43 | 89.45 | 91.46 | 69.95 | 93.59 | 71.16 | 95.69 | 81.00 |
| | 10 | 85.52 | 96.67 | 93.56 | 60.49 | 90.47 | 80.23 | 83.53 | 96.92 | 91.68 | 95.38 | 55.11 | 95.87 | 76.16 | 94.00 | 93.41 | 89.43 | 91.45 | 69.98 | 93.57 | 71.39 | 95.67 | 80.95 |
| | 25 | 85.56 | 96.67 | 93.57 | 60.37 | 90.47 | 80.88 | 83.50 | 96.90 | 91.66 | 95.38 | 55.32 | 95.86 | 76.25 | 93.98 | 93.38 | 89.41 | 91.44 | 69.99 | 93.53 | 71.55 | 95.66 | 80.92 |
| $e$FW | 5 | 84.55 | 96.33 | 94.88 | 55.66 | 90.74 | 75.54 | 83.63 | 95.58 | 89.60 | 94.71 | 54.33 | 95.93 | 75.78 | 93.84 | 93.14 | 91.21 | 91.02 | 69.56 | 92.96 | 69.87 | 90.60 | 80.65 |
| | 10 | 84.60 | 96.34 | 94.88 | 55.66 | 90.73 | 76.53 | 83.63 | 95.58 | 89.58 | 94.70 | 54.33 | 95.97 | 75.81 | 93.84 | 93.14 | 91.22 | 91.02 | 69.54 | 93.02 | 69.89 | 90.60 | 80.66 |
| | 25 | 84.55 | 96.33 | 94.88 | 55.66 | 90.73 | 75.47 | 83.63 | 95.58 | 89.60 | 94.70 | 54.33 | 95.97 | 75.81 | 93.84 | 93.14 | 91.22 | 91.02 | 69.54 | 93.02 | 69.89 | 90.60 | 80.66 |

**Table 9:** Fined-grained results on PASCAL VOC validation set.

### F.4.1 Results for longer inference regime

We show in Figure 2 a comparison of the discrete energy across the methods on a subset of 10 *val* images of PASCAL VOC for 100 inference iterations, using DeepLabv3+ and Potts dense CRF.

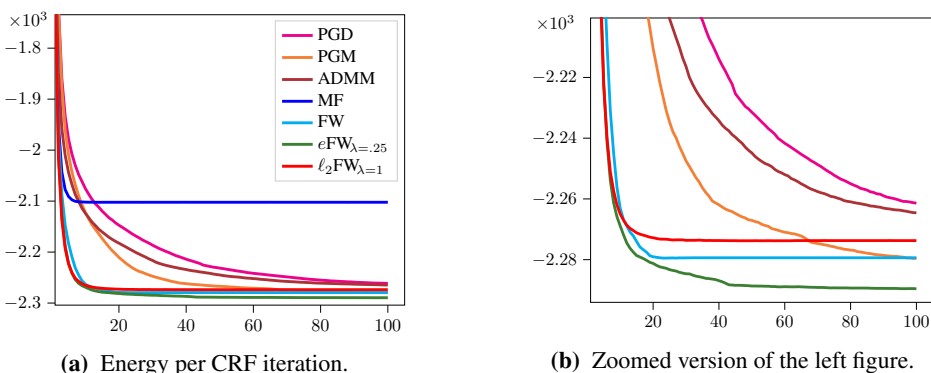

**(a)** Energy per CRF iteration.  **(b)** Zoomed version of the left figure.

**Figure 2: CRF energy** averaged over a subset of 10 *val* images of **PASCAL VOC** using DeepLabv3+ and Potts dense CRF.

From these results, we observe that:

1. Vanilla FW and $\ell_2$FW already converge after around 20 iterations. $\ell_2$FW does better than vanilla FW only in the early iterations.

2. PGM surpasses $\ell_2$FW at after 70 iterations, and surpasses vanilla FW after 100 iterations.

3. PGD and ADMM are likely to surpass $\ell_2$FW and vanilla, too, if given sufficient number of iterations, as these do not show any sign of convergence yet.

The main observation here is that, the relative performance of the methods are different between the early (typically first 10 iterations) and the later stage. In §6 we gave some hypotheses on why the proposed regularized Frank-Wolfe may work better than the others. Our main argument is that vanilla Frank-Wolfe is already much better than the other methods (in the first few iterations), and what we do is to equip it with the ability of effectively learning with SGD (potential improvements in terms of energy are rather a byproduct and not the main objective, as the improvements are sometimes small). Let us summarize this situation as follows:

1. Vanilla Frank-Wolfe outperforms other first-order methods such as PGD, PGM, and ADMM **during the first few iterations** (and may be surpassed at a later stage, as already shown).

2. For SGD learning, in which only a small number of iterations (due to the vanishing/exploding gradient problems, as already observed in previous work [76]), this behavior (reaching quickly a very low energy) of vanilla Frank-Wolfe is highly desirable.

3. Unfortunately, vanilla Frank-Wolfe iterates are piecewise constant and thus the resulting gradients are zero almost everywhere, which makes learning through backpropagation impossible.

4. Our regularized Frank-Wolfe is designed to precisely solve this zero-gradient issue.

### F.4.2 Additional inference results

Figures 3 and 4 show more results for the inference experiments.

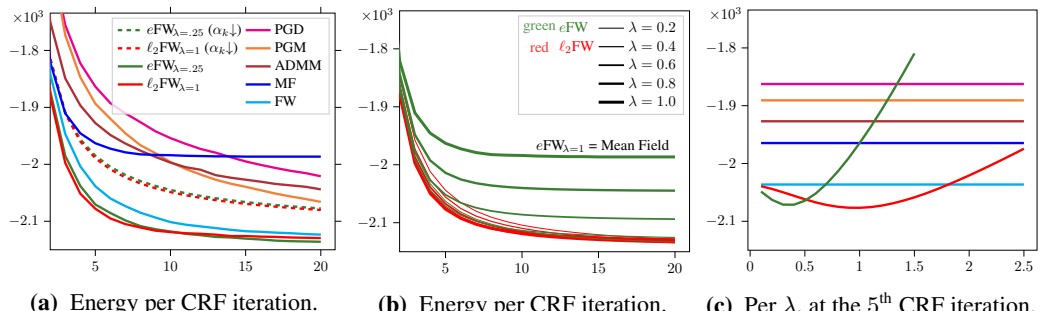

**(a)** Energy per CRF iteration.     **(b)** Energy per CRF iteration.     **(c)** Per $\lambda$, at the $5^{\text{th}}$ CRF iteration.

**Figure 3: CRF energy** averaged over 1449 *val* images of **PASCAL VOC** using DeepLabv3+ and Potts dense CRF. **(a)** Comparison between regularized Frank-Wolfe and the other methods for some selected values of the regularization weight $\lambda$. **(b)** Results of regularized Frank-Wolfe for different values of $\lambda$. **(c)** Energy per $\lambda$ after 5 iterations. Best viewed in color.

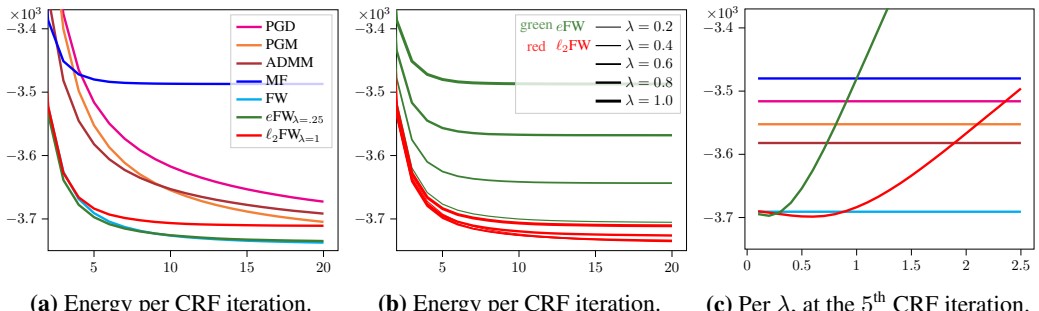

**(a)** Energy per CRF iteration.     **(b)** Energy per CRF iteration.     **(c)** Per $\lambda$, at the $5^{\text{th}}$ CRF iteration.

**Figure 4: CRF energy** averaged over 500 *val* images of **Cityscapes** using DeepLabv3+ and Potts dense CRF. **(a)** Comparison between regularized Frank-Wolfe and the other methods for some selected values of the regularization weight $\lambda$. **(b)** Results of regularized Frank-Wolfe for different values of $\lambda$. **(c)** Energy per $\lambda$ after 5 iterations. Best viewed in color.