# OpenReview forum: "Regularized Frank-Wolfe for Dense CRFs: Generalizing Mean Field and Beyond"
_NeurIPS.cc/2021/Conference — NeurIPS 2021 Poster_

### Official Review · Reviewer_HFqk · 2021-07-15

**Rating:** 7
**Confidence:** 4

**Summary:**

This paper presents a general algorithm for CRF learning and inference.  The algorithm is a regularized version of the classical FW algorithm.  It generalizes several existing algorithms , such as mean field and the concave-convex procedure.  Empirical studies on semantic segmentation tasks demonstrate intriguing properties of the proposed regularized FW algorithm.

**Limitations And Societal Impact:**

No.

For the semantic segmentation task, could discuss potential  threat to user privacy when it is used for surveillance tasks.


**Main Review:**

Contributions:

The authors presents a generic FW algorithm:  regularized FW, and there are various FW variants that can be viewed as special cases of the regularized FW.  In terms of originality,  it is more appropriate to be stated as a summarization of FW variants and giving unified analysis.



Clarity and related work:

The presentation is clear and well-organized, the main technical details are well stated.  It is a pleasure to read the paper.

Regarding mean field algorithms, recently it is shown that classical mean field problems  enjoys  provable guarantees with submodular/supermodular energies sovled by the DR-DoubleGreedy algorithm [1], it would be
interesting to see whether the regularized FW has some guanrantees/advantages for this class of problems:

[1] Bian, Y., Buhmann, J., & Krause, A. (2019, May). Optimal continuous dr-submodular maximization and applications to provable mean field inference. In International Conference on Machine Learning (pp. 644-653). PMLR.


Experiments:

The authors conducted extensive experimental evaluations againgst several baselines, such as PGD, ADMM, and MF,  with two variants of FW:  l2-FW and eFW.  They found that dense CRFs can achieve improvements over the strong DeepLabv3+ CNN baselines.  the best variant of regularized FW achieves an mIoU score of 88 on the PASCAL VOC test set.  The experimental evaluations are solid, in my point of view.




**Time Spent Reviewing:**

6.5

---

> ### Author Response · Authors · 2021-08-10
> **Responses to Reviewer HFqk (or R3)**
>
> We thank the reviewer for their detailed feedback and for their appreciation of the paper. We have updated the manuscript to include the improvements suggested by the reviewer, which is detailed below.
>
> > Regarding mean field algorithms, recently it is shown that classical mean field problems enjoys provable guarantees with submodular/supermodular energies sovled by the DR-DoubleGreedy algorithm [1], it would be interesting to see whether the regularized FW has some guanrantees/advantages for this class of problems:
> [1] Bian, Y., Buhmann, J., & Krause, A. (2019, May). Optimal continuous dr-submodular maximization and applications to provable mean field inference. In International Conference on Machine Learning (pp. 644-653). PMLR.
>
> We thank the reviewer for this reference. After spending some time on this problem, we found that if the energy satisfies some conditions that are weaker than being submodular, then there is only a unique mean field fixed point (this fact can be proved using some results from Winston and Kolter, Monotone operator equilibrium networks (NeurIPS 2020)). On the other hand, it can be shown that a point is a mean field fixed point if and only if it is a stationary point of our entropy-regularized energy, consequently the unique mean field fixed point is necessarily the global optimum and our algorithm is guaranteed to converge to this point (with suitable stepsizes).
>
> On the other hand, we have not figured out how to proceed with supermodular energies. This is an interesting avenue for future work. We have modified the paper to reflect this, as shown in this [screenshot](https://i.imgur.com/zAwczyt.png) (where [10] is the suggested Bian et al. reference).
>
> > **Limitations And Societal Impact** For the semantic segmentation task, could discuss potential threat to user privacy when it is used for surveillance tasks.
>
> We thank the reviewer for pointing this out. We have added the following paragraph to the end of the manuscript:
>
> > **Societal impact** Semantic segmentation models can be used in surveillance systems, which might raise potential privacy concerns. Furthermore, the datasets that our models were trained on are known to present strong built-in bias [64], thus they should be used with caution.
> >
> > [64] Antonio Torralba and Alexei A. Efros. Unbiased look at dataset bias. In CVPR, 2011.
>
> ([screenshot](https://i.imgur.com/pbO2gKn.png))
>
> ---
>
> We thank the reviewer again for their feedback. Please feel free to let us know if you have any remaining concerns or questions, we will be happy to address them.

---

### Official Review · Reviewer_xtoq · 2021-07-16

**Rating:** 7
**Confidence:** 4

**Summary:**

Semantic segmentation is a widely-studied problem in computer vision, and over the years a range of techniques have been popular. This paper re-visits the use of dense CRF layers on top of a neural network, which can be used to model pairwise connectivity across output labels. They contribute a new optimization method for mean field inference in the CRF that converges quickly (provably), and is differentiable (such that the whole model can be trained end-to-end). The paper pushes the SOTA on two popular segmentation benchmarks.

**Limitations And Societal Impact:**

The authors write N/A for "Did you discuss any potential negative societal impacts of your work?"

I understand the thought process: they are contributing core algorithms for a task that is well-established. However, it would have been helpful if the authors commented on the potential biases (e.g., geographical) in this benchmarking dataset.

**Main Review:**

 **originality**
The authors draw on sophisticated optimization techniques (regularized Frank-Wolfe) that have not been applied in the context of mean-field CRF inference. Care needs to be taken because the optimization technique also needs to be differentiable, so that the model can be learned end-to-end.

**clarity/quality**

The writing is generally high-quality and the paper's contributions are placed carefully in the context of related work.  The paper carefully reproduces recently-published models on a popular computer vision benchmark using the standard evaluation setup.

I don't have technical background to evaluate the theoretical statements about convergence rates in Sec 4. Hopefully another reviewer does, and I defer to their judgment.

I am familiar with the CRF literature, and found the exposition sound.

**significance**
Dense CRFs were popular for semantic segmentation, but have fallen out of interest in recent years. This paper combines dense CRFs with new feature extraction networks and a new way to do CRF inference.The authors hope that this will spark renewed interest in dense CRFs. With that in mind, the paper is significant, as it provides SOTA results and could help the research dialog pivot towards new methods.

My primary hesitation (see below) is that I am unsure if the magnitude of improvement on this task is characteristic of papers that get published based on pushing SOTA. Hopefully the authors can clarify in their response.

**magnitude of accuracy improvement**

I'm unfamiliar with the recent literature on semantic segmentation. Is the magnitude of performance increase in Table 3 characteristic of recently-published papers? How do you know that this difference is statistically significant?

**related work**
The idea of using FW to do marginal inference was established in earlier work. See, for example, "Bethe Learning of Conditional Random Fields via MAP Decoding" and " New outer bounds on the marginal polytope"

Can you comment on the relationship between this and your work?

**end-to-end learning**
Could you learn details of the FW algorithm (e.g., the step size schedule) end-to-end as well?

**vanishing/exploding gradients**
I didn't follow the argument in L377 about vanishing/exploding gradients. Since each update of FW adds something to the current iterate, it resembles a residual network, which is known to suffer less from these issues. Why is unrolled FW different? Is the issue that the unrolled optimization is unstable (e.g., the step size is too big)?

**test-time usage**
What happens when you train with 5 iterations, but use more (e.g., 25) at test time?

**zero gradients**

L336 suggests that the projection step  is problematic, since it yields zero gradients. However, doesn't this contradict 354, where you use \ell_2 because it performs the best?

**fine-grained analysis**
I would appreciate some anecdotes regarding cases where your method seems to meaningfully improve performance over the baseline that doesn't use a CRF layer. Is there some story to where the CRF is useful?


**Time Spent Reviewing:**

2

---

> ### Author Response · Authors · 2021-08-10
> **Responses to Reviewer xtoq (or R2) [1/2]**
>
> **[1/2]**
>
> We thank the reviewer for their detailed feedback. We respond to their questions below.
>
> > **significance** My primary hesitation (see below) is that I am unsure if the magnitude of improvement on this task is characteristic of papers that get published based on pushing SOTA. Hopefully the authors can clarify in their response.
>
> We would like to clarify that the goal of our paper is not to simply push SoTA on semantic segmentation. It is **a theoretical paper with encouraging practical results**, and we believe that this is already clear from the presentation, e.g., in the introduction, our primary contributions are presented in L21-L36; moreover, the majority of the paper presents theoretical and algorithmic results.
>
> Even though our results on PASCAL VOC (88% mIoU) could be considered SoTA in the standard settings (i.e., no external data except COCO for pre-training), we did not focus on this in the presentation. As seen in the paper, we were very careful and conservative when discussing our experimental results using the (modest) adjective "encouraging".
>
> We hope that our response has clarified things for the reviewer, and also hope that our theoretical and algorithmic contributions are appreciated as much as (if not more than) the empirical results.
>
> > **magnitude of accuracy improvement** I'm unfamiliar with the recent literature on semantic segmentation. Is the magnitude of performance increase in Table 3 characteristic of recently-published papers? How do you know that this difference is statistically significant?
>
> These questions appear to arise due to the assumption that the goal of this paper is to push SoTA on semantic segmentation. Hopefully our previous response has clarified this and changed the reviewer’s opinion of the paper. However, in case the reviewer is still concerned with the magnitudes of improvement on the test sets (especially on Cityscapes this is smaller, as we already acknowledged in the paper), please see the following.
>
> Let us consider the following results from DeepLabv3 and DeepLabv3+ for example: [[DeepLab results on test sets]](https://i.imgur.com/zkrzN6X.jpg).
> We take DeepLabv3 and DeepLabv3+ as they are the most relevant to our work, but there are many similar examples.
>
> These models were considered SoTA at the time of their publication. The tables above show that they offered an improvement of 0.1% mIoU over the best existing models. Note that DeepLabv3 and DeepLabv3+ were pre-trained with the huge JFT-300M dataset, which brought an extra 0.8%-1.2% (please see the papers for details). Now imagine that these papers were written by a non-Google researcher, and because they didn't have access to JFT-300M which is an in-house dataset of Google, the results were 1.1% below prior art. Would this have downgraded any merits of these papers? That should not be the case, because the technical contributions of the two versions (JFT vs non-JFT) are almost identical.
>
> As the reviewer may be aware already, published results on leaderboards often require a lot of tricks to achieve those reported numbers. For example, in the DeepLabv3 paper we note the following: [[DeepLab tricks for test sets]](https://i.imgur.com/jv8yYjn.png). This is also used in DeepLabv3+, which employs the same training protocol as DeepLabv3. Such tricks are completely orthogonal to the main technical contributions of these papers, which was proposing new architectures. Therefore, if the authors did not use them and lost an extra 1%, their technical contributions would have remained almost the same and *should* have been appreciated almost equally by a fair reader.
>
> The main point here is that "high magnitude of accuracy improvement over existing work" should not play a major role in the judgment of a paper. We consider it only as the cherry on the cake. Having the cherry would be nice, but the cake itself could be very good even without the cherry!
>
> For our paper, we could have obtained a higher improvement (on the test sets) over DeepLabv3+ using more "tricks". For example, the authors of CRFasRNN (Zheng et al., 2015) performed a grid-search over the kernel width and weight parameters of the CRF (i.e., variables $\alpha,\beta,\gamma,w_1,w_2$ in Eq. (108), Section B.2 of our appendix) before end-to-end fine tuning with a very small learning rate. Note that doing a grid-search over a 5-dimensional space is an expensive task (even if it is replaced with a more efficient hyper-parameter tuning method, such as Bayesian optimization), and would make the training pipeline more complex. We did not do this because our results already supported our hypotheses. Some further improvements would have been possible certainly, but would have offered only marginal added value to the paper.
>
> We would be happy to discuss this point further with the reviewer.
>
> > **related work** The idea of using FW to do marginal inference was established in earlier work. See, for example, "Bethe Learning of Conditional Random Fields via MAP Decoding" and "New outer bounds on the marginal polytope". Can you comment on the relationship between this and your work?
>
> We thank the reviewer for the references. In our paper, we cited Meshi et al. (2015), Desmaison et al. (2016), and Lê-Huu and Paragios (2018) for having used Frank-Wolfe for inference, but indeed we also missed other prior work. The algorithms by Tang et al. (2016) and Sontag and Jaakkola (2007) are similar to Meshi et al. (2015) (they differ in the used regularizers). Interestingly, all these algorithms are also special cases of our framework. We have revised the manuscript to include this discussion (please see: [screenshot1](https://i.imgur.com/AkIt8bj.png) and [screenshot2](https://i.imgur.com/reQbeYr.png)).
>
> > **end-to-end learning** Could you learn details of the FW algorithm (e.g., the step size schedule) end-to-end as well?
>
> This is indeed an interesting ablation study that we could include in the paper. We carried out an additional experiment with L2-FW and Entropy-FW ($\lambda = 0.7$) in which we allowed the stepsize $\alpha_k$ at each CRF iteration to be learnable (initialized at $0.5$). We observed that:
>
> - The stepsizes at all the steps behave very similarly.
> - For Entropy-FW, the stepsizes increased during training, but for L2-FW, they decreased.
>
> In addition, we also tried allowing the regularization weight $\lambda$ to be learnable as well. We initialized it at $1.0$ for L2-FW and at $0.7$ for Entropy-FW. For both models, we found that $\lambda$ increased during training.
>
> Please refer to this [TensorBoard screenshot](https://i.imgur.com/CzDVAm6.jpg) for visual results. (We apologize for providing TensorBoard plots instead of those with the same format as in the manuscript, because performing the additional experiments took a lot of time and in the end we could not be able to put these results properly into the manuscript yet.)
>
> Regarding the accuracy, we did not observe significant differences compared to fixed $\alpha_k$ and fixed $\lambda$. There are a few things that we could try, for example setting a different learning rate for these parameters (different from the learning rate of the CRF potential parameters), but we believe that this kind of fine-grained analysis would be more suitable for a future extended journal version of our work.
>
> We will revise the manuscript to include the above discussion.
>
> > **vanishing/exploding gradients** I didn't follow the argument in L377 about vanishing/exploding gradients. Since each update of FW adds something to the current iterate, it resembles a residual network, which is known to suffer less from these issues. Why is unrolled FW different? Is the issue that the unrolled optimization is unstable (e.g., the step size is too big)?
>
> “Each update of FW adds something to the current iterate, it resembles a residual network” is actually an interesting observation! It should be noted that in all the algorithms we consider (including mean field and ADMM/PGD/PGM), there is already a residual connection from the output $u$ of the CNN to each iterate $x^k$ (this is because $x^k$ is a function of the gradient $\nabla E(x^{k-1}) = Px^{k-1}+u$). In regularized FW, there are additional connections between the iterates (if the stepsize is lower than 1), but at the same time there are also extra nodes ($p^k$) in the computation graph, so the two might cancel out. An in-depth analysis of the gradient flow would be necessary to draw any conclusions, but this is out of scope of the paper.
>
> With respect to L377, the vanishing/exploding gradient issue was actually not attributed to FW by that argument, but rather to the other methods. L374-L377 can be read informally as follows: “FW algorithms achieve low energies in the early stage, and other first-order methods (ADMM/PGD/PGM) may potentially catch up at a later stage (around 100 iterations), but using 100 iterations would cause vanishing/exploding gradients” ← this implicitly attributes the vanishing/exploding gradient problem to ADMM/PGD/PGM, because there’s no reason to run FW for such a large number of iterations.
>
> To make our argument clearer and more convincing while keeping the paper’s focus, we have updated the text as follows: [screenshot](https://i.imgur.com/H67IS3w.png).
>
> ----
> (We are running out of allowed characters, please see the second part [2/2])

---

> > ### Comment · Reviewer_xtoq · 2021-08-18
> > **thanks for the detailed response**
> >
> > I appreciate all of the clarifications, particularly regarding SOTA and the evaluation protocols. I agree with your perspective. I have increased my review to accept.

---

> > > ### Author Response · Authors · 2021-08-19
> > > **Thank you for your feedback**
> > >
> > > We are happy that our responses are appreciated. We thank the reviewer again for the feedback and for having increased the rating!

---

> ### Author Response · Authors · 2021-08-10
> **Responses to Reviewer xtoq (or R2) [2/2]**
>
> **[2/2]**
>
> > **zero gradients** L336 suggests that the projection step is problematic, since it yields zero gradients. However, doesn't this contradict 354, where you use \ell_2 because it performs the best?
>
> L336 says that the projection step is differentiable almost everywhere and not fully differentiable, which does not mean “zero gradients”. Please note that the zero-gradient issue occurs when the update is piecewise constant (such as in vanilla Frank-Wolfe). The projection step in L2-FW does not have this problem (although because it is only differentiable almost everywhere, training can be unstable for large learning rates, as noted in the paper: L332-L337), otherwise its learning curve would have been a horizontal line, as shown in Figure 3 for vanilla Frank-Wolfe. The reason one obtains a horizontal line in the case of zero gradients is that SGD performs its updates as $\theta = \theta - \eta*\nabla f(\theta)$, thus if $\nabla f(\theta) = 0$ then it stays where it is forever.
>
> We have added the following clarification in the footnote on page 3: [[screenshot]](https://i.imgur.com/yfbvq6n.png).
>
> > **test-time usage** What happens when you train with 5 iterations, but use more (e.g., 25) at test time?
> >
> > **fine-grained analysis** I would appreciate some anecdotes regarding cases where your method seems to meaningfully improve performance over the baseline that doesn't use a CRF layer. Is there some story to where the CRF is useful?
>
> We thank the reviewer for these questions. Indeed, at inference time one can increase the number of CRF iterations to potentially improve the results. We did not do that to simplify the evaluation protocol (due to the large number of experiments).
>
> To give an answer, we carried out an additional experiment on PASCAL VOC. We randomly picked a trained model (from Table 2) for DeepLabv3+-L2-FW and DeepLabv3+-Entropy-FW ($\lambda=0.7$) and evaluated them using 5, 10, and 25 CRF iterations on the validation set. We obtain the following results: [[Fine-grained Results]](https://i.imgur.com/8NwgjDp.png), which show that using more iterations could improve the results but only slightly.
>
> Regarding the suggested fine-grained analysis, also from the above results, we observe that:
> - L2-FW can improve significantly (>= 2% mIoU) over CNN on the following classes: bicycle (+15.67%), boat (+8.74%), bus (2.26%), chair (2.56%), dining table (5.86%), sofa (4.21%), train (3.37%), tvmonitor (2.54%).
> - L2-FW is slightly worse than CNN on the following classes: cat (-0.04%)
> - L2-FW can improve slightly (<2%) over CNN on the remaining classes.
> - Entropy-FW can improve significantly (>= 2% mIoU) over CNN on the following classes: aeroplane (3.08%), bicycle (+10.78%), boat (+4.05%), bus (2.26%), dining table (5.53%), motorbike (2.95%), sofa (2.93%), tvmonitor (2.20%).
> - Entropy-FW is slightly worse than CNN on the following classes: car (-1.94%), cat (-0.71%), train (-1.73%)
> - Entropy-FW can improve slightly (<2%) over CNN on the remaining classes.
>
> We can see that there are a lot of consensus between the two solvers (though there are also differences, especially on the car class).
>
> We thank the reviewer again for the suggestion and will revise the manuscript to include these ablation studies.
>
> > **Limitations And Societal Impact:**
> The authors write N/A for "Did you discuss any potential negative societal impacts of your work?" I understand the thought process: they are contributing core algorithms for a task that is well-established. However, it would have been helpful if the authors commented on the potential biases (e.g., geographical) in this benchmarking dataset.
>
> We thank the reviewer for pointing this out. We have added the following paragraph to the end of the manuscript:
>
> > **Societal impact** Semantic segmentation models can be used in surveillance systems, which might raise potential privacy concerns. Furthermore, the datasets that our models were trained on are known to present strong built-in bias [64], thus they should be used with caution.
> >
> > [64] Antonio Torralba and Alexei A. Efros. Unbiased look at dataset bias. In CVPR, 2011.
>
> [[screenshot]](https://i.imgur.com/pbO2gKn.png)
>
> ---
>
> We thank the reviewer again for their feedback. We would be happy to address any remaining concerns or questions.

---

### Official Review · Reviewer_nMc7 · 2021-07-20

**Rating:** 7
**Confidence:** 4

**Summary:**

The authors propose a new generalized FW strategy that captures many existing approximate MAP strategies as a special case.  They demonstrate that this new approach is easy to implement and can yield improvements over a CNN alone using a joint CRF/CNN framework for image segmentation.

**Limitations And Societal Impact:**

Yes, more remains to be done, but the work is a promising start.

**Main Review:**

Originality:  While the key ideas are based primarily on existing work, this work does make several key contributions related to rates of convergence and accuracy of rounding that are worthwhile in this specific context.

Quality:  1) My only criticism is the somewhat limited experiments.  I think that the one experiment does indeed showcase the strengths (and some of the weakness) of the proposed approach, but as the strategy is widely applicable, it would have been nice to see a more diverse set of experiments.  Still, the paper is already jam-packed, so I don't consider it a significant drawback.

2) There is a lot of work on FW for approximate inference / learning that isn't cited here (of course it would be tough to get it all).  One citation that comes to mind is:

Kui Tang, Nicholas Ruozzi, David Belanger, and Tony Jebara. Bethe learning of graphical models via MAP decoding. Nineteenth International Conference on Artificial Intelligence and Statistics (AISTATS), May 2016.

3) Some comparison of constant versus diminishing stepsize would be interesting in the experimental section.

Clarity:  The paper is well-written and easy to follow (some of this is due to most of the technical proofs being relegated to the appendix).  My only complaint: All of your figures need axis labels.

Significance:  New and improved approximate MAP / learning algorithms promise to reinvigorate works that have, at least until recently, explored the benefits of combined approaches.

**Time Spent Reviewing:**

3 hours

---

> ### Author Response · Authors · 2021-08-10
> **Responses to Reviewer nMc7 (or R1)**
>
> We thank the reviewer for their detailed feedback and for their appreciation of the paper. We have updated the manuscript to include the improvements suggested by the reviewer, which is detailed below together with our responses to their questions.
>
> > 1. My only criticism is the somewhat limited experiments. I think that the one experiment does indeed showcase the strengths (and some of the weakness) of the proposed approach, but as the strategy is widely applicable, it would have been nice to see a more diverse set of experiments. Still, the paper is already jam-packed, so I don't consider it a significant drawback.
>
> We thank the reviewer for the feedback. Given the amount of *theoretical* and *algorithmic* contributions we presented in the paper, we had to make a trade-off on the *application* side. We chose dense CRFs because they are challenging (the full-connectivity makes a large number of existing inference methods intractable) and motivating (as framed in the paper's Introduction). This choice naturally led to semantic segmentation for the experiments as dense CRFs are known for their success on this task.
>
> Even with this conservative choice, the paper is already jam-packed, as the reviewer noticed. We would also like to point out that for this single task, we considered 2 CNN backbones, 7 CRF solvers, 2 datasets, and 2 experimental settings (inference vs end-to-end learning). We acknowledge, however, that this is still a minor limitation of the paper (as the reviewer also noted), which we hope to address in a future extended journal version of our work.
>
>
> > 2. There is a lot of work on FW for approximate inference / learning that isn't cited here (of course it would be tough to get it all). One citation that comes to mind is:
> Kui Tang, Nicholas Ruozzi, David Belanger, and Tony Jebara. Bethe learning of graphical models via MAP decoding. Nineteenth International Conference on Artificial Intelligence and Statistics (AISTATS), May 2016.
>
> We thank the reviewer for the reference. In our paper, we cited Meshi et al. (2015), Desmaison et al. (2016), and Lê-Huu and Paragios (2018) for having used Frank-Wolfe for inference, but indeed we missed some other prior work. We have added Tang et al. (2016) and also Sontag and Jaakkola (2007) (New Outer Bounds on the Marginal Polytope) to the revised manuscript (please see: [screenshot1](https://i.imgur.com/AkIt8bj.png) and [screenshot2](https://i.imgur.com/reQbeYr.png)). Interestingly, these algorithms are similar to Meshi et al. (2015) and are also special cases of our framework.
>
>
> > 3. Some comparison of constant versus diminishing stepsize would be interesting in the experimental section.
>
> Although not presented in the main content, we empirically found that the constant stepsize $\alpha_k = 1$ produced the lowest energies -- we briefly discussed this in Appendix B.1. There is infact a theoretical justification for this. We used a dense CRF model with pairwise Gaussian potentials, which has a nice property: the resulting energy function is approximately concave (see (Krahenbuhl and Koltun, 2013, Section 3.2) for details). When the energy is concave, the constant step-size $\alpha_k = 1$ is optimal, i.e., the energy decreases the most at each iteration. This result is discussed in L255 and can be obtained from Theorem 1 and Table 5 in the appendix.
>
> In the revised manuscript, we will add the above discussion and will add results for the standard stepsizes $\alpha_k = \frac{k}{k+2}$ to Figure 1a. Please see this [screenshot3](https://i.imgur.com/qbAwjub.png).
>
> > Clarity: My only complaint: All of your figures need axis labels.
>
> Thank you for pointing this out. In the submitted manuscript we moved the axis labels to the legends (as "$y$ per $x$", e.g., "energy per iteration" or "mIoU per epoch"). We will add the axis labels in the revised manuscript.
>
> ---
> We thank the reviewer again for their feedback. Please feel free to let us know if you have any remaining concerns or questions, we will be happy to address them.

---

> > ### Comment · Reviewer_nMc7 · 2021-09-01
> > **Thanks!**
> >
> > Thanks for your feedback.

---

### Public Comment · Authors · 2021-12-08
**Poster session**

Dear reviewers, area chairs, and readers,

We are currently presenting the paper at Spot A0 in NeurIPS virtual world. Please visit and have a chat with us!

Best regards.

---
NeurIPS landing page: https://neurips.cc/virtual/2021/poster/26419

Poster session: https://eventhosts.gather.town/app/J94QBQeiWGjDI7sQ/3-optimization

---

### Decision · Program_Chairs · 2021-09-27

**Decision:**

Accept (Poster)

**Comment:**

The paper is well written and the experiments are convincing. The authors spent a lot of effort addressing reviewer comments and all reviewers increased their score to 7. I would therefore like to accept the paper.

At first, generalized Frank-Wolfe seems like an odd choice for solving (6). Indeed, (6) can be solved by PG or MD, which have a better convergence rate than generalized FW. Moreover, since adding L2 or entropic regularization leads to Euclidean or KL projections instead of usual LMOs, the proposed algorithm is quite similar to PG and MD. I would therefore encourage the authors to further justify their choice. One justification is the connection with parallel mean field, provided that it doesn't hold for MD. In my understanding, another justification would be that FW-like step sizes require no tuning, which is important for backpropagability. In contrast, PG or MD would require either knowledge of the Lipschitz constant or a line search.

A few of minor comments on the technical side:

  * The rates are not state of the art. The authors mention a 1/\sqrt{t} rate for the "adaptive" step-size, but a O(1/t) rate is known for that class of step-sizes, see for instance http://proceedings.mlr.press/v54/locatello17a/locatello17a.pdf and/or https://arxiv.org/pdf/1806.05123.pdf

  * The analysis of the generalized FW algorithm with a strongly convex regularizer is very similar (although not fully identical) to the analysis with strongly convex set. I encourage the authors to mention and/or compare against this line of work, see for instance https://arxiv.org/pdf/2011.03351.pdf

  * The adaptive step-size still requires to know constants that can be difficult to compute in practice or crude upper bounds. This should be clarified.

* Generalized FW algorithms are also proposed in

F. Bach. Duality between subgradient and conditional gradient methods. SIAM Journal of Optimization